# Combined transcranial magnetic stimulation and electroencephalography reveals alterations in cortical excitability during pain

Nahian Shahmat Chowdhury[1,2]*, Alan KI Chiang[1,2], Samantha K Millard[1,2], Patrick Skippen[1], Wei-Ju Chang[1,3], David A Seminowicz[1,4], Siobhan M Schabrun[1,5]

[1]Center for Pain IMPACT, Neuroscience Research Australia, Sydney, Australia; [2]University of New South Wales, Sydney, Australia; [3]School of Health Sciences, College of Health, Medicine and Wellbeing, The University of Newcastle, Callaghan, Australia; [4]Department of Medical Biophysics, Schulich School of Medicine & Dentistry, University of Western Ontario, London, Canada; [5]The Gray Centre for Mobility and Activity, University of Western Ontario, London, Canada

**Abstract** Transcranial magnetic stimulation (TMS) has been used to examine inhibitory and facilitatory circuits during experimental pain and in chronic pain populations. However, current applications of TMS to pain have been restricted to measurements of motor evoked potentials (MEPs) from peripheral muscles. Here, TMS was combined with electroencephalography (EEG) to determine whether experimental pain could induce alterations in cortical inhibitory/facilitatory activity observed in TMS-evoked potentials (TEPs). In Experiment 1 (n=29), multiple sustained thermal stimuli were administered to the forearm, with the first, second, and third block of thermal stimuli consisting of warm but non-painful (pre-pain block), painful (pain block) and warm but non-painful (post-pain block) temperatures, respectively. During each stimulus, TMS pulses were delivered while EEG (64 channels) was simultaneously recorded. Verbal pain ratings were collected between TMS pulses. Relative to pre-pain warm stimuli, painful stimuli led to an increase in the amplitude of the fronto-central negative peak ~45 ms post-TMS (N45), with a larger increase associated with higher pain ratings. Experiments 2 and 3 (n=10 in each) showed that the increase in the N45 in response to pain was not due to changes in sensory potentials associated with TMS, or a result of stronger reafferent muscle feedback during pain. This is the first study to use combined TMS-EEG to examine alterations in cortical excitability in response to pain. These results suggest that the N45 TEP peak, which indexes GABAergic neurotransmission, is implicated in pain perception and is a potential marker of individual differences in pain sensitivity.

*For correspondence:
n.chowdhury@neura.edu.au

Competing interest: The authors declare that no competing interests exist.

## eLife assessment

This **valuable** study provides **convincing** evidence that acute experimental pain induces changes of cortical excitability. Although the modality specificity of the findings is not fully clear, the findings will be of interest to researchers interested in the brain mechanisms of pain.

## Introduction

Pain is a complex subjective experience, and understanding how pain is processed remains a challenge (*Apkarian, 2021*). Several neuroimaging techniques have been applied to disentangle these

**Figure 1.** Schematic of experimental apparatus. The apparatus consisted of transcranial magnetic stimulation (TMS) during concurrent electroencephalography (EEG) to simultaneously record motor-evoked potentials (MEPs) and TMS-evoked potentials (TEPs). MEPs were recorded using electromyographic (EMG) electrodes placed over the distal region of the extensor carpi radialis brevis (ECRB), while thermal pain was delivered over the proximal region of the ECRB.

complexities: functional magnetic resonance imaging has assisted in identifying brain structures implicated in pain processing (*Reddan and Wager, 2018*), while electroencephalography (EEG) has contributed to our understanding of the temporal sequence of pain processing (*Ploner and May, 2018*). Another useful technique that has been used to examine the role of inhibitory and facilitatory neural circuits in pain has been transcranial magnetic stimulation (TMS) delivered to the brain (*Chang et al., 2018*; *Schabrun and Hodges, 2012*). However, current applications of TMS to pain have involved recording the output of TMS from a muscle, a signal that could be influenced by many intermediate (subcortical, spinal, peripheral) factors, and which restricts investigations to the motor system only. Here, we used combined TMS-EEG measure to record output of TMS directly from the cortex and from multiple brain regions, in pain-free and tonic pain conditions.

When TMS is delivered over the primary motor cortex (M1), a magnetic pulse induces an electrical current in underlying cortical tissue that, if the intensity is sufficient, activates corticomotor pathways, inducing a motor evoked potential (MEP) in a target muscle. The magnitude of the MEP serves as an index of corticomotor excitability. Past systematic reviews on studies measuring MEPs during acute experimental pain (*Bank et al., 2013*; *Burns et al., 2016*; *Chowdhury et al., 2022a*; *Rohel et al., 2021*) have shown a reduction in MEP amplitude during pain and after pain resolution, with stronger reductions in MEP amplitude associated with lower acute pain severity (*Chowdhury et al., 2022a*). It has been hypothesized that this reduction in MEP amplitude is an adaptive mechanism that restricts movement in the pain-afflicted area, to protect the area from further pain and injury (*Hodges and Tucker, 2011*).

While previous findings show promise for the use of TMS to discover and validate potential biomarkers for pain, limitations exist when using TMS to measure MEPs. First, MEP responses to TMS reflect the net sum of cortical, spinal, and peripheral activity within the corticomotor pathway. This makes it unclear as to whether pain processes occur at the cortical, spinal, or peripheral level. Further, measurement of MEPs restricts investigations to M1. One way of overcoming these limitations is by combining TMS and EEG to measure TMS-evoked potentials (TEPs). TEPs index cortical excitability *directly* from the cortex (i.e. without influence of subcortical, spinal, and peripheral processes), as well as from regions *outside M1* (*Farzan et al., 2016*). TEPs also provide an index of the activity of specific neurotransmitter circuits within the cortex. For example, TEP peaks that occur at ~45 ms and 100 ms post-stimulation are linked to GABA$_A$ and GABA$_B$ neurotransmission, respectively (*Premoli et al., 2014*), whereas the TEP peak ~60 ms post-stimulation is linked to glutamatergic neurotransmission

(*Belardinelli et al., 2021*). Overall, TEPs provides additional spatial and temporal information about cortical activity over MEPs, making it ideal for understanding the brain mechanisms involved in pain perception.

TEPs have already shown potential to serve as a biomarker for the development and prognosis of various neurological and psychiatric conditions for reviews see (*Kallioniemi and Daskalakis, 2022*; *Tremblay et al., 2019*). However, the applicability of TEPs to pain research is yet to be established. While GABAergic processes indexed by TEPs have been hypothesized to be involved in pain (*Barr et al., 2013*), direct evidence is scarce. Two studies (*Che et al., 2019*; *Ye et al., 2022*) examined whether the potential analgesic effects of repetitive TMS (rTMS) over the dorsal prefrontal cortex are associated with plasticity in TEPs. These studies separately measured TEPs and ratings to painful stimuli, before and after rTMS, with one finding that increases in pain thresholds following rTMS were associated with changes in TEPs that index GABAergic processes (*Ye et al., 2022*). While these studies assist us in understanding whether TEPs might mediate rTMS-induced pain reductions, no study has investigated whether TEPs are altered in direct response to pain.

The aim of the present study was to use TMS-EEG to determine whether acute experimental pain induces alterations in cortical inhibitory and facilitatory peaks observed using TEPs. We used a tonic heat pain paradigm (*Furman et al., 2020*; *Granot et al., 2006*), in which multiple thermal stimuli were applied over the right extensor carpi radialis brevis (ECRB) muscle via a thermode. For each thermal stimulus, the temperature increased from a neutral baseline of 32 °C to either a warm non-painful or a painful (46 °C) temperature, with this temperature maintained for 40 s. During this time, TMS was administered to the left M1 with concurrent EEG to obtain TEPs from 63 scalp channels, and MEPs from the ECRB muscle (see *Figure 1*). Verbal pain ratings were obtained between pulses. It was hypothesized that TEP peaks that index GABAergic processes, including the peaks at ~45 and 100 ms after TMS, would increase in response to painful stimuli relative to warm non-painful stimuli.

## Results

### Experiment 1 – Does acute pain alter cortical excitability?

#### Design

In Experiment 1 (n=29), we determined whether painful thermal stimuli induced alterations in TEP peaks relative to a non-painful baseline. The protocol (*Figure 2*) consisted of three blocks of stimuli, in chronological order: pre-pain, pain, and post-pain blocks. The pre-pain and post-pain blocks each consisted of six 40 s thermal stimuli (20 s interstimulus interval) delivered at a non-painful temperature (calibrated to each participant's warmth detection threshold), while the pain block consisted of six 40 s thermal stimuli delivered at 46 °C. The pre-pain/pain/post-pain design has been commonly used in the TMS-MEP pain literature, as many studies have demonstrated strong changes in corticomotor excitability that persist beyond the painful period. Indeed, in a systematic review, we showed effect

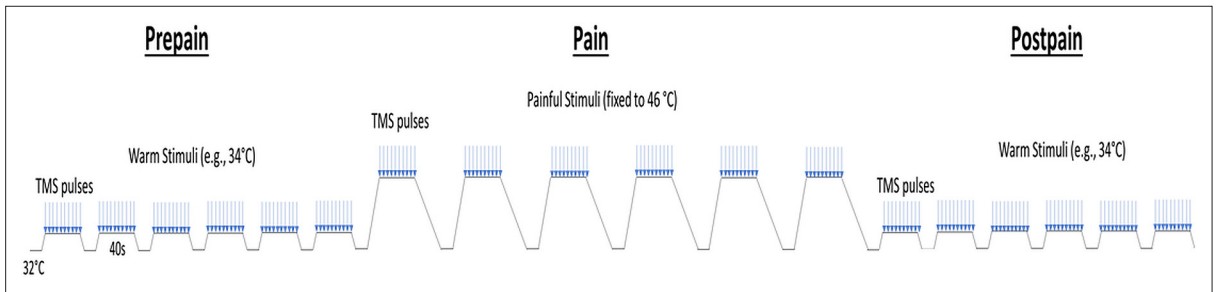

**Figure 2.** Schematic of the protocol for Experiment 1. Participants experienced three blocks of thermal stimuli: a pre-pain, pain, and post-pain block, with each block consisting of multiple thermal stimuli delivered 40 s at a time, and during which TMS measurements (indicated by blue arrows) and verbal pain ratings were obtained. The pre-pain and post-pain blocks involved thermal stimuli delivered at the warm threshold (i.e. the temperature that leads to any perceived change in skin temperature from baseline). In the pain block, thermal stimuli were delivered at 46 °C.

The online version of this article includes the following figure supplement(s) for figure 2:

**Figure supplement 1.** Schematic of the protocol for Experiment 2.

**Figure supplement 2.** Schematic of the protocol for Experiment 3.

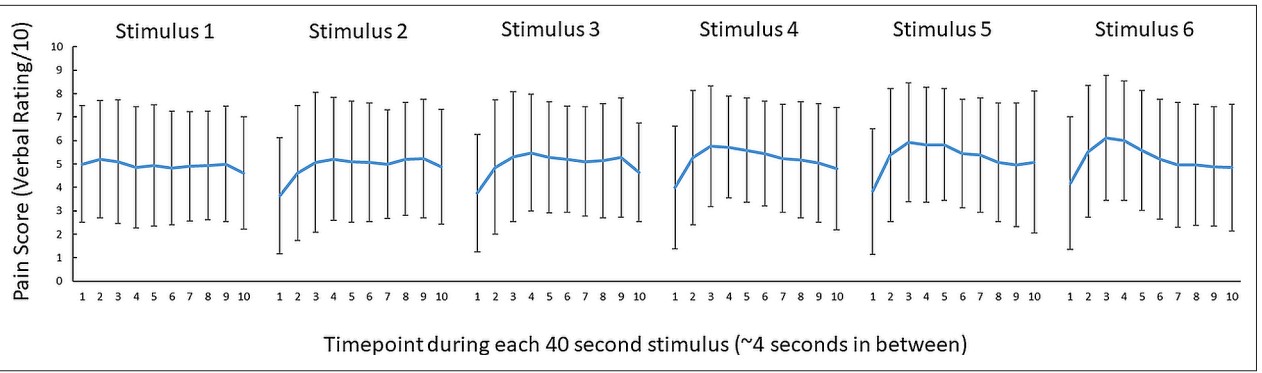

**Figure 3.** No conclusive evidence of a difference in pain ratings between successive 46 °C 40 s thermal stimuli. Mean (± SD) pain ratings (n = 29) during the 6 thermal stimuli delivered during the pain block (thermal stimuli delivered at 46 °C) of Experiment 1. Ten pain ratings were collected over each 40-s thermal stimulus ~every 4 s. A 6 (stimulus number: 1–6) x 10 (timepoint:1–10) Bayesian repeated measures ANOVA revealed anecdotal evidence (i.e. no conclusive evidence) of a difference in pain between six thermal stimuli ($BF_{10}$=2.86), very strong evidence for a difference in pain ratings between the 10 timepoints ($BF_{10}$=6.1$^{30}$) and strong evidence of an interaction between stimulus number and timepoint ($BF_{10}$=19.6).

The online version of this article includes the following figure supplement(s) for figure 3:

**Figure supplement 1.** Pain ratings for Experiment 2.

**Figure supplement 2.** Warmth ratings for Experiment 3.

**Figure supplement 3.** Pain ratings for Experiment 3.

sizes of 0.55–0.9 for MEP reductions 0–30 min after pain had resolved (*Chowdhury et al., 2022a*). As such, if we had used an alternative design with blocks of warm stimuli intermixed with blocks of painful stimuli, the warm stimuli blocks would not serve as a valid non-painful baseline. Based on a previous study (*Dubé and Mercier, 2011*) which also used sequences of painful (50 °C) and warm (36 °C) thermal stimuli, we did not anticipate that the stimulus in the pain block would entrain pain in the post-pain block.

## Quantitative sensory testing

Prior to the test blocks, we measured warmth, cool, and pain detection thresholds to ascertain whether: (a) participants could perceive increases or decreases in the thermode temperature relative to a neutral baseline of 32 °C, and (b) the pain detection threshold was below 46 °C. All participants were able to detect increases or decreases in temperature from baseline. The mean (± SD) cool and warmth detection threshold was 28.6 ± 1.9°C and 35.1 ± 1.5°C, respectively. All participants reported a heat pain threshold that was above their warmth detection threshold and below the test temperature of 46 °C. The mean heat pain threshold was 41.2 ± 2.8°C.

## Pain ratings

All participants reported 0/10 pain during the pre-pain and post-pain blocks, and pain ratings varying between 1 and 10 during the pain block. *Figure 3* shows the mean pain ratings for the 10 pain measurements of each of the 6 painful stimuli delivered during the pain block (~4 s in between pain measurements). A 6 (stimulus number: 1–6) x 10 (timepoint:1–10) Bayesian repeated measures ANOVA revealed anecdotal evidence (i.e. no conclusive evidence) of a difference in pain between six thermal stimuli ($BF_{10}$=2.86). However, there was very strong evidence for a difference in pain ratings between the 10 timepoints ($BF_{10}$=6.1$^{30}$). There was also strong evidence of an interaction between stimulus number and timepoint, suggesting the time course of pain across the 40 s thermal stimulus differed across the six thermal stimuli of the pain block ($BF_{10}$=19.6). Overall, although there was no conclusive evidence for pain differing *between* successive stimuli, there was evidence that pain fluctuated *during* each 40-s stimulus.

## Motor-evoked potentials

The mean resting motor threshold (RMT) and test intensity of TMS was (mean ± SD) 70.7 ± 8.5% and 77.7 ± 9.2% of maximum stimulus output respectively. We note that the relatively high RMTs are

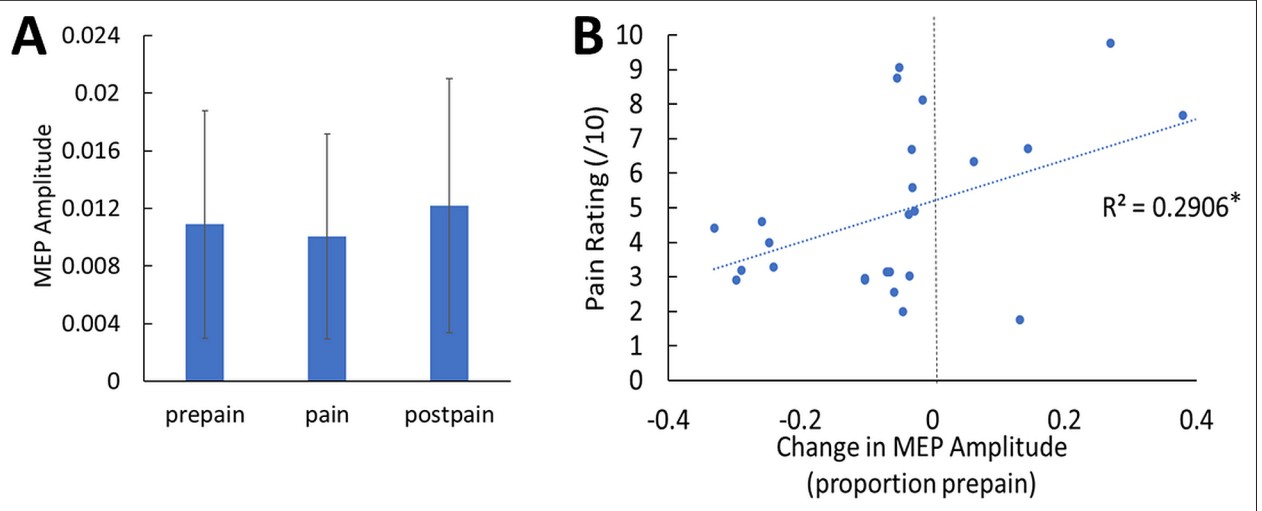

**Figure 4.** No conclusive evidence of MEP amplitude differences between conditions; however individual pain sensitivity was predicted by changes in MEP amplitude. (**A**) Mean (± SD) MEP amplitude (n=26) during the pre-pain, pain, and post-pain blocks of Experiment 1. A Bayesian repeated-measures ANOVA revealed anecdotal evidence of a difference in MEP amplitude between blocks ($BF_{10}$=1.02). (**B**) Individual-level Relationship between change in MEP amplitude during pain (proportion of pre-pain) and mean verbal pain rating provided by each participant. There was strong evidence for a positive relationship ($r_{26}$=0.54, $BF_{10}$=11.17).

likely due to aspects of the experimental setup that increased the distance between the TMS coil and the scalp, including the layer of foam placed over the coil, the EEG cap and relatively thick electrodes (6 mm). Three participants were excluded from the MEP analysis due to EMG software failure – these participants were still included in the TEP analysis. A Bayesian repeated-measures ANOVA was run to compare MEP amplitudes between pre-pain, pain, and post-pain conditions. There was anecdotal evidence of a difference in MEP amplitude between blocks ($BF_{10}$=1.02; *Figure 4A*). A Bayesian correlation test was also run to determine whether the mean pain rating (across blocks and timepoints) was associated with the change in MEP amplitude during pain as a proportion of pre-pain. These MEP change values were log-transformed as they were not normally distributed according to a Shapiro-Wilk Test ($W$=0.58, p<0.001). There was strong evidence for a positive relationship ($r_{26}$=0.54, $BF_{10}$=11.17; *Figure 4B*). such that participants who showed a larger reduction in MEP amplitude during pain reported lower pain ratings.

## TMS-evoked potentials

One participant was excluded from the TEP analysis due to failure to save the recording during the experiment, though this participant was still included in the MEP analysis. One participant had missing post-pain data as the TMS coil had overheated during this portion of the experiment – their data were still included for the pre-pain vs. pain comparison. *Figure 5* shows the grand average TEPs for all 63 channels, across pre-pain, pain, and post-pain conditions, as well as the scalp topographies at timepoints where TEP peaks are typically observed – N15, P30, N45, P60, N100, and P180ms (*Farzan et al., 2016*). Source reconstruction using a co-registered template brain model was also conducted to characterize source activity at each timepoint (*Figure 5*). For the N15 and P30 peaks, there was higher current density in the left motor areas, consistent with previous studies suggesting that TMS evoked activity at 15 and 30ms after the TMS pulse reflect early excitation of motor areas ipsilateral to the stimulated region (*Farzan and Bortoletto, 2022*). For the N45 and P60 peaks, there was higher current density in the left motor and somatosensory areas at 45 and 60ms after the TMS pulse, consistent with previous studies showing a sensorimotor origin at these timepoints (*Ahn and Fröhlich, 2021*); however, higher current density was also present in the left parietal and right sensorimotor areas. For the N100 and P180 peaks, there was higher current density in the central regions, mostly contralateral to the stimulated cortex. Overall, we found consistencies in the source localization with previous studies, including a sensorimotor origin of early peaks from 15 to 60ms. However, we did not directly compare source activity between conditions due to the inaccuracies involved in source

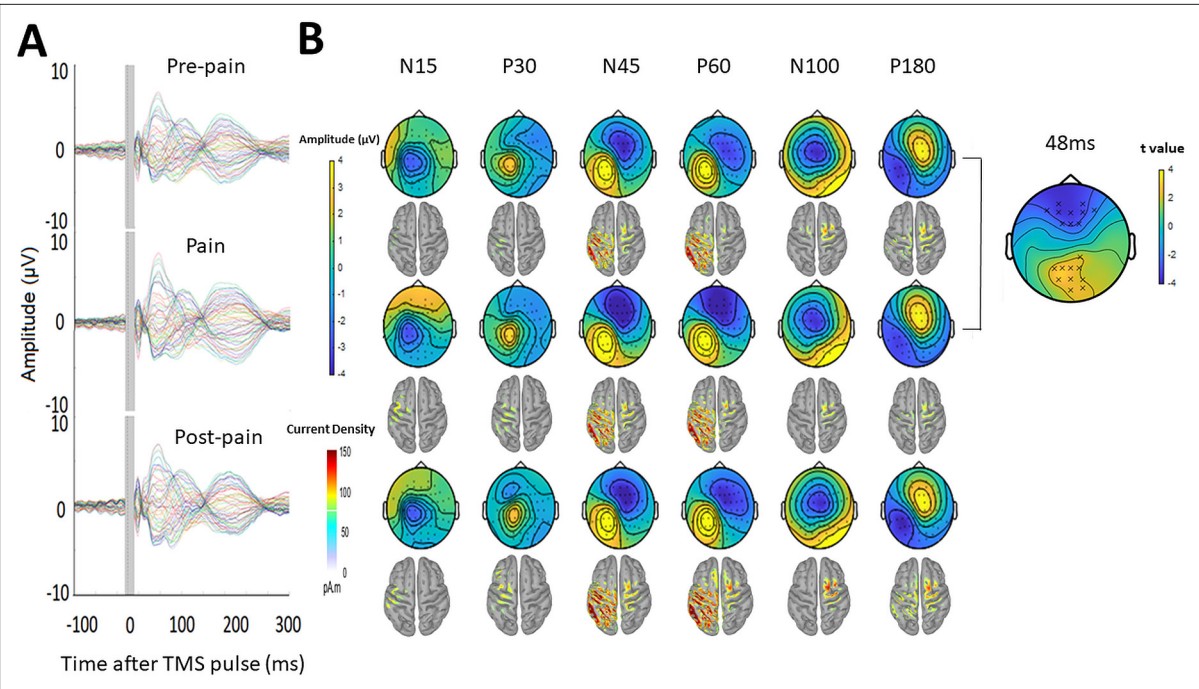

**Figure 5.** Pain led to increased negative and positive amplitude in frontocentral and parietal-occipital sites respectively, 43–90ms after the TMS pulse. (**A**) Grand-average TEPs (n = 28) during the pre-pain, pain, and post-pain blocks of Experiment 1. The gray-shaded area represents the window of interpolation around the TMS pulse. (**B**) Scalp topographies and estimated source activity at timepoints where TEP peaks are commonly observed, including the N15, P30, N45, P60, N100, and P180. A cluster plot is also shown on the right comparing signal amplitude between the pain and pre-pain conditions at a representative timepoint (48ms) between 43 and 90ms, which is where significant amplitude differences were observed. The black stars demonstrate the presence of significant positive (yellow) clusters or negative (blue) clusters.

estimation in the absence of co-registered magnetic resonance imaging (MRI) scans (**Brodbeck et al., 2011**; **Michel and Brunet, 2019**) and EEG electrode location digitization (**Shirazi and Huang, 2019**).

Comparisons of TEP amplitude between conditions were based on the electrode-space data. However, as this is the first study investigating the effects of experimental pain on TEP amplitude, there were no a priori regions or timepoints of interest to compare between conditions. A statistically robust starting point in these situations is to use a cluster-based permutation analysis (**Frömer et al., 2018**). This analysis was used to compare amplitudes between pre-pain and pain, and pre-pain and post-pain, at each timepoint and for each electrode. We found that during pain relative to pre-pain, there was a significantly larger negative amplitude (p=0.021) at frontocentral electrodes and a significantly larger positive amplitude at parietal-occipital electrodes (p=0.028), specifically between 43 and 90ms after the TMS pulse. No significant differences in TEP amplitude were found when comparing the pre-pain and post-pain conditions, and pain and post-pain conditions. As such, the subsequent TEP peak analyses were focused on the pre-pain vs. pain comparison, while the pre-pain vs. post-pain and pain vs. postpain comparisons are presented in the figure supplements.

*Figure 6A* shows the grand average TEP waveform at the frontocentral electrodes ('AF3','AFz','AF 4','F1','Fz','F2','F4','FC2','FC4') identified from the cluster analysis for the pre-pain vs. pain conditions (*Figure 6—figure supplement 1* shows the post-pain comparisons). Two peaks at ~45 and 85 ms after the TMS pulse are visible in the time window where the significant cluster was detected. Given the approximate timing, these peaks are likely to be the N45 peak and an early N100 peak. The amplitude of these peaks was identified for each participant using the TESA peak function (**Rogasch et al., 2017**) with defined time windows of 40–70 and 75–95ms for the first and second peak respectively. These time windows were chosen to account for variation between participants in the latency of the first and second peak. Bayesian paired-sample t-tests showed very strong evidence that the first peak at ~45 ms ($BF_{10}$=57.21) and moderate evidence that the second peak at ~85ms ($BF_{10}$=6.77) had larger amplitude during the pain block compared to the pre-pain block. *Figure 6B* shows the individual level relationship between the mean pain rating, and the difference in N45 and N100 amplitude between

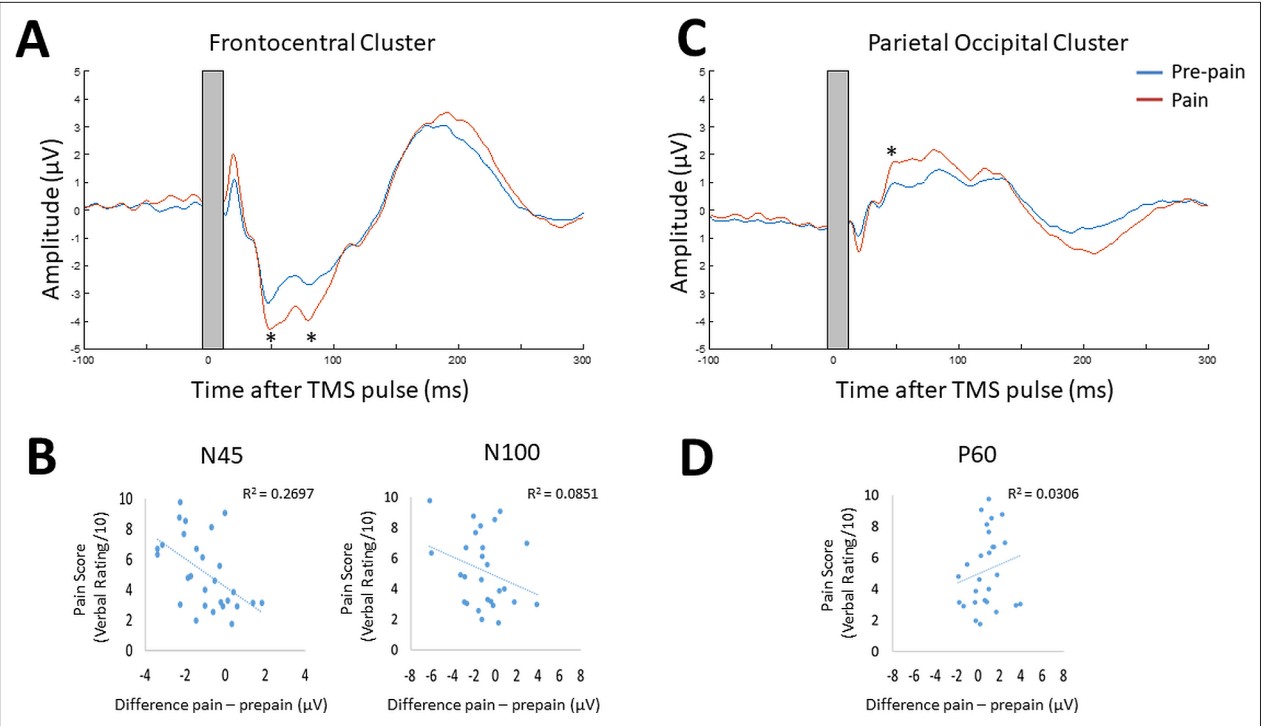

**Figure 6.** Pain led to increases in N45, P60, and N100 peak amplitude, and individual pain sensitivity was predicted by changes in the N45 peak. TEPs (n = 28) across pain and pre-pain condition for the frontocentral electrodes (**A**) and parietal-occipital electrodes (**C**) identified from the cluster analysis of Experiment 1. The grey shaded area represents the window of interpolation around the transcranial magnetic stimulation (TMS) pulse. For the frontocentral electrodes, there was evidence for stronger negative peaks at ~45 and 85ms post-TMS. For the parietal-occipital electrodes, there was evidence for a stronger positive peak was identified at ~50ms post-TMS. The astericks indicates at least moderate evidence for the alternative hypothesis that the peak amplitude is larger in pain vs. pre-pain (BF$_{10}$ >3). Individual-level relationship between mean verbal pain ratings provided by each participant and change in peak amplitudes at ~45ms (N45),~85ms (N100) post-TMS (**B**), and ~50ms (P60) post-TMS (**D**). The astericks indicates at least moderate evidence for a relationship between change in peak amplitude, and verbal pain ratings (BF$_{10}$ >3).

The online version of this article includes the following figure supplement(s) for figure 6:

**Figure supplement 1.** No conclusive evidence of a difference in TEP peak amplitude between post-pain and pre-pain and pain conditions.

pain and pre-pain. There was strong evidence that participants who reported higher pain ratings also showed a larger increase in N45 peak amplitude during the pain block ($r_{26}$=0.52, BF$_{10}$=10.64). There was anecdotal evidence for no association between pain ratings and changes in the N100 peak amplitude during pain ($r_{26}$=0.24, BF$_{10}$=0.48).

*Figure 6C* shows the mean TEP waveform of the parietal-occipital electrodes ('P1','PO3','O1','CPz','Pz','Pz','Oz','CP2','P2', 'PO4','O2','CP4','P4') identified from the cluster analysis for the pre-pain vs. pain conditions (*Figure 6—figure supplement 1* shows the post-pain comparisons). One peak at ~50ms is visible in the time window where the significant cluster was detected. The approximate timing of this peak is consistent with the commonly identified P60. The amplitude of this peak was identified for each participant with a defined time window of 35–65ms. This time window was chosen to account for variation between participants in the latency of the peak. There was moderate evidence that the peak at ~50ms was stronger during the pain block compared to pre-pain block (BF$_{10}$=5.56). *Figure 6D* shows the individual level relationship between the mean pain rating and the difference in the P60 amplitude between pain and pre-pain. There was anecdotal evidence in favour of no relationship between pain ratings and changes in P60 amplitude during the pain block ($r_{26}$=0.21, BF$_{10}$=0.407).

## Relationship between changes in TEP peaks and MEP amplitude

Respectively, there was moderate and anecdotal evidence for no relationship between alterations in MEP amplitude and alterations in the N100 ($r_{25}$=−.14, BF$_{10}$=0.303) and P60 ($r_{25}$=0.185, BF$_{10}$=0.36) during pain. There was anecdotal evidence for a relationship between alterations in the N45 and MEP amplitude during pain ($r_{25}$=−.387, BF$_{10}$=1.40).

## Experiment 2 – Does acute pain alter cortical excitability or sensory potentials?

### Design

Several studies have shown that a significant portion of TEPs do not reflect the direct cortical response to TMS, but rather auditory potentials elicited by the 'clicking' sound from the TMS coil, and somatosensory potentials elicited by the 'flicking' sensation on the skin of the scalp (*Biabani et al., 2019*; *Chowdhury et al., 2022b*; *Conde et al., 2019*; *Rocchi et al., 2021*). Indeed, the signal at ~100ms post-TMS from Experiment 1 may reflect an auditory N100 response. As it is extremely challenging to isolate and filter these auditory- and somatosensory-evoked potentials using pre-processing pipelines, (*Ilmoniemi and Kičić, 2010*; *Massimini et al., 2005*). However, recent studies have shown that even when these methods are used, sensory contamination of TEPs is still present, as shown by commonalities in the signal between active and sensory sham conditions that mimic the auditory/somatosensory aspects of real TMS (*Biabani et al., 2019*; *Conde et al., 2019*; *Rocchi et al., 2021*). This has led many leading authors (*Biabani et al., 2019*; *Conde et al., 2019*) to recommend the use of sham conditions to control for sensory contamination. To separate the direct cortical response to TMS from sensory evoked activity, Experiment 2 (n=10) included a sham TMS condition that mimicked the auditory/somatosensory aspects of active TMS to determine whether any alterations in the TEP peaks in response to pain were due to changes in sensory evoked activity associated with TMS, as opposed to changes in cortical excitability. A similar design (*Figure 2—figure supplement 1*) was used to Experiment 1, with the inclusion of a sham TMS condition within the pre-pain and pain blocks, and exclusion of the post-pain block, since the aim was to identify the source of the pain effect from Experiment 1. The sham TMS condition was similar to a recent study (*Gordon et al., 2021*), involving the delivery of the TMS coil rotated 90° to the scalp to simulate the auditory component associated with real TMS, and concurrent electrical

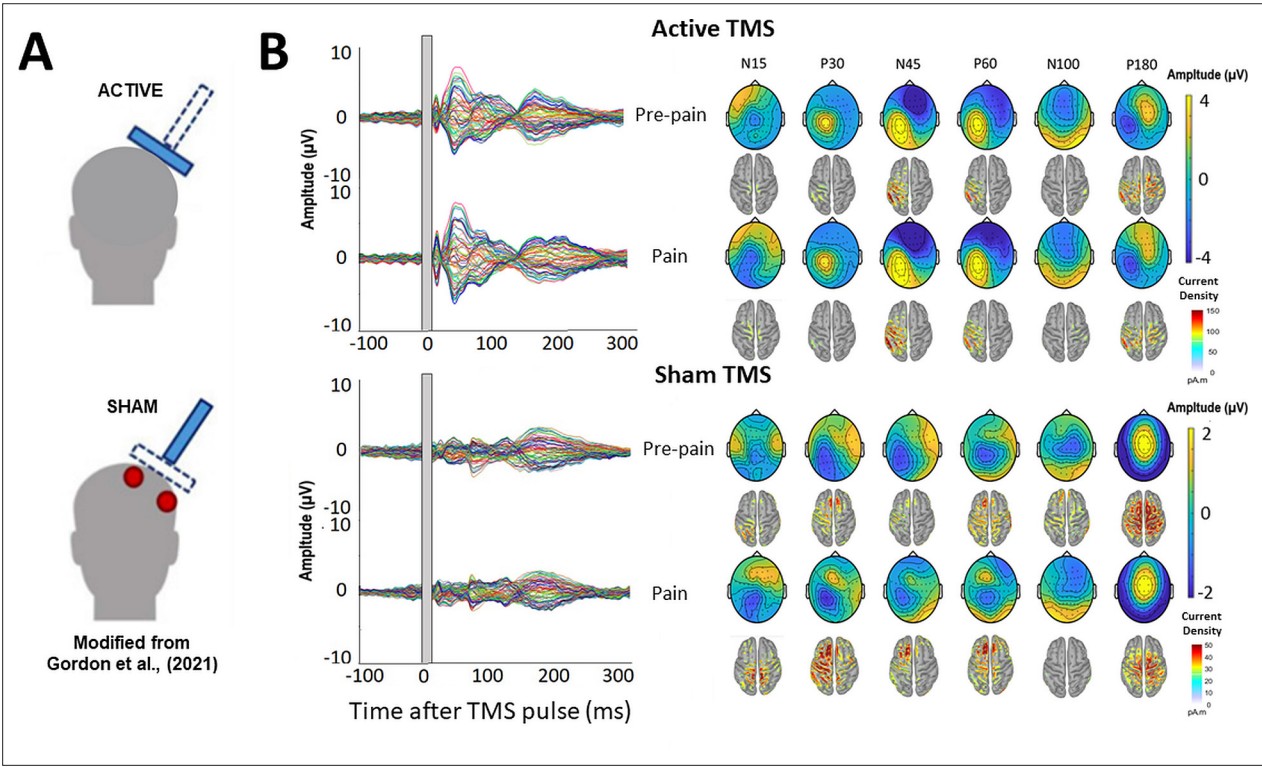

**Figure 7.** TMS-evoked potentials for Active and Sham TMS. (**A**) Schematics showing the delivery of active and sham TMS of Experiment 2. Sham TMS involved scalp electrical stimulation (in red) beneath a sham coil (in dotted blue) to mimic somatosensory stimulation associated with active TMS, and concurrent delivery of active TMS 90° to the scalp (in shaded blue) to mimic auditory stimulation associated with TMS. (**B**) Left: TEPs (n = 10) during the pre-pain and pain blocks, for both active and sham stimulation. The grey shaded area represents the window of interpolation around the TMS pulse. Right: Scalp topographies and estimated source activity at timepoints where TEP peaks are commonly observed, including the N15, P30, N45, P60, N100, and P180.

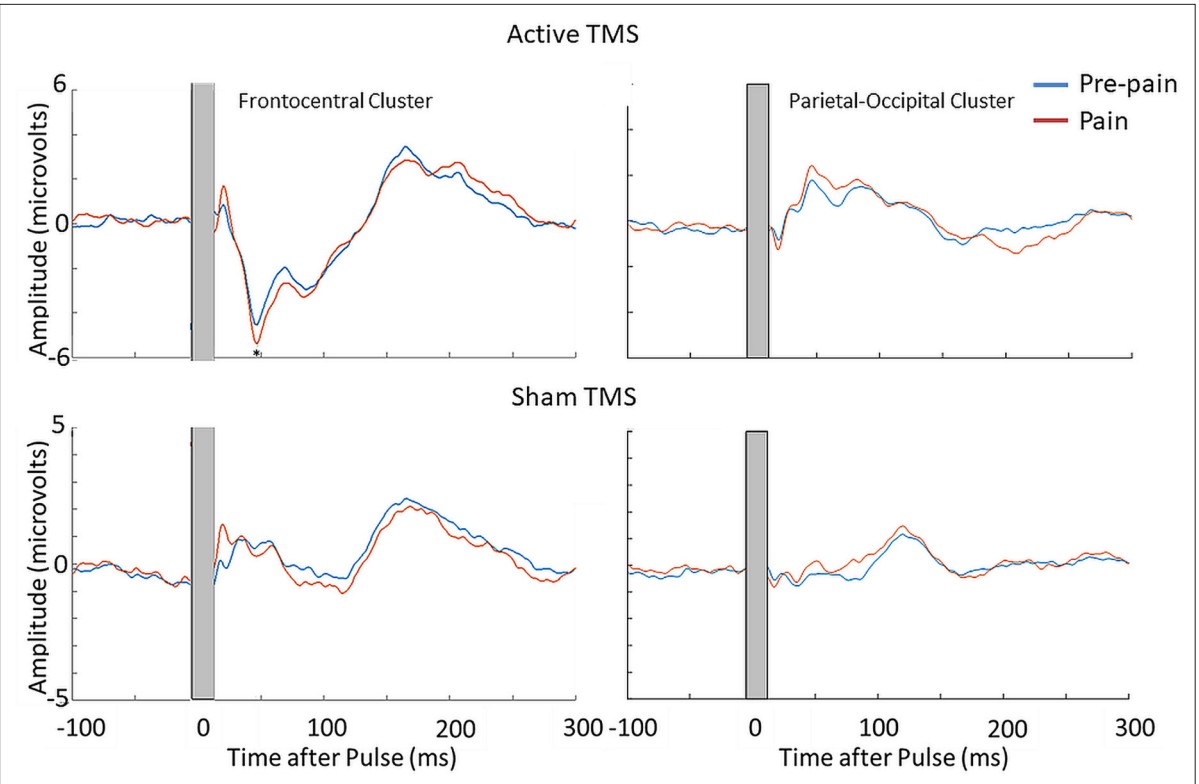

**Figure 8.** Pain led to an increase in the N45 peak amplitude during active TMS but not sham TMS. TEPs (n = 10) during pain and pre-pain blocks, across active and sham TMS conditions of Experiment 2, for the frontocentral electrodes (left) and parietal-occipital electrodes (right) identified from the cluster analysis in the main experiment. A significantly stronger frontocentral negative peak was identified ~45ms post-TMS during pain compared to pre-pain, for the active TMS condition. The astericks indicates at least moderate evidence for the alternative hypothesis that the amplitude is larger in pain vs. pre-pain (BF$_{10}$ >3).

stimulation beneath a sham coil to simulate the somatosensory component associated with real TMS (see *Figure 7A*).

## Pain ratings

The mean cold and warm detection threshold was 29.6 ± 2.6°C and 36.0 ± 3.6°C, respectively. The mean heat pain threshold was 42.1 ± 3.6 °C. *Figure 3—figure supplement 1* shows the pain ratings during the pain block for each condition.

## TMS-evoked potentials

The mean RMT and test stimulus intensity was, respectively, 70.4 ± 4.3% and 77.7 ± 4.7% of maximum stimulus output. The mean test electrical stimulation intensity for the sham TMS condition was 4.4±2.5 mA. This intensity is comparable to previous studies using sham electrical stimulation and is insufficient to directly activate the cortex (*Chowdhury et al., 2022b*; *Conde et al., 2019*; *Rocchi et al., 2021*). *Figure 7B* shows the grand average TEPs, scalp topographies and estimated source activity for active and sham TMS, across pre-pain and pain conditions. *Figure 8* shows the mean TEP waveform for the frontocentral and parietal-occipital clusters identified from Experiment 1, across active and sham conditions. There was moderate evidence that the amplitude of the N45 peak was increased during pain vs. pre-pain blocks for active TMS (BF$_{10}$=3.26), and moderate evidence for no difference between pain and pre-pain blocks for sham TMS (BF$_{10}$=0.309). When comparing pain and pre-pain blocks, there was, respectively, moderate and anecdotal evidence for no alterations in the frontocentral N100 for active (BF$_{10}$=0.31) and sham TMS (BF$_{10}$=0.42). There was anecdotal evidence for no alteration in the parietal occipital P60 for both active (BF$_{10}$=0.786) and sham TMS (BF$_{10}$=0.42).

Overall, the results showed that the N45 peak was altered in response to pain for active but not sham TMS, suggesting the experience of pain led to an alteration in the excitability of the cortex, and not the auditory/somatosensory aspects of TMS.

## Experiment 3 – Does acute pain alter cortical excitability or reafferent muscle activity?

### Design

Previous studies have shown that a significant portion of the TEP peaks at 45 and 60ms post-TMS reflect reafferent feedback from the muscle twitch in response to suprathreshold TMS applied over M1. This comes from MRI-informed EEG studies showing source localization of the N45 and P60 peaks to the somatosensory areas, as well as correlations between MEP amplitude and N45/P60 amplitude (*Ahn and Fröhlich, 2021*; *Petrichella et al., 2017*). Indeed, Experiments 1 and 2 also showed localization of the N45 and P60 to sensorimotor areas. As such, Experiment 3 recruited a further ten participants to determine whether the pain-induced increase in the N45 peak was due to stronger reafferent feedback from muscle twitches. A design (*Figure 2—figure supplement 2*) similar to Experiment 1 was used, with the inclusion of a subthreshold TMS condition (90% RMT) within the pre-pain and pain blocks.

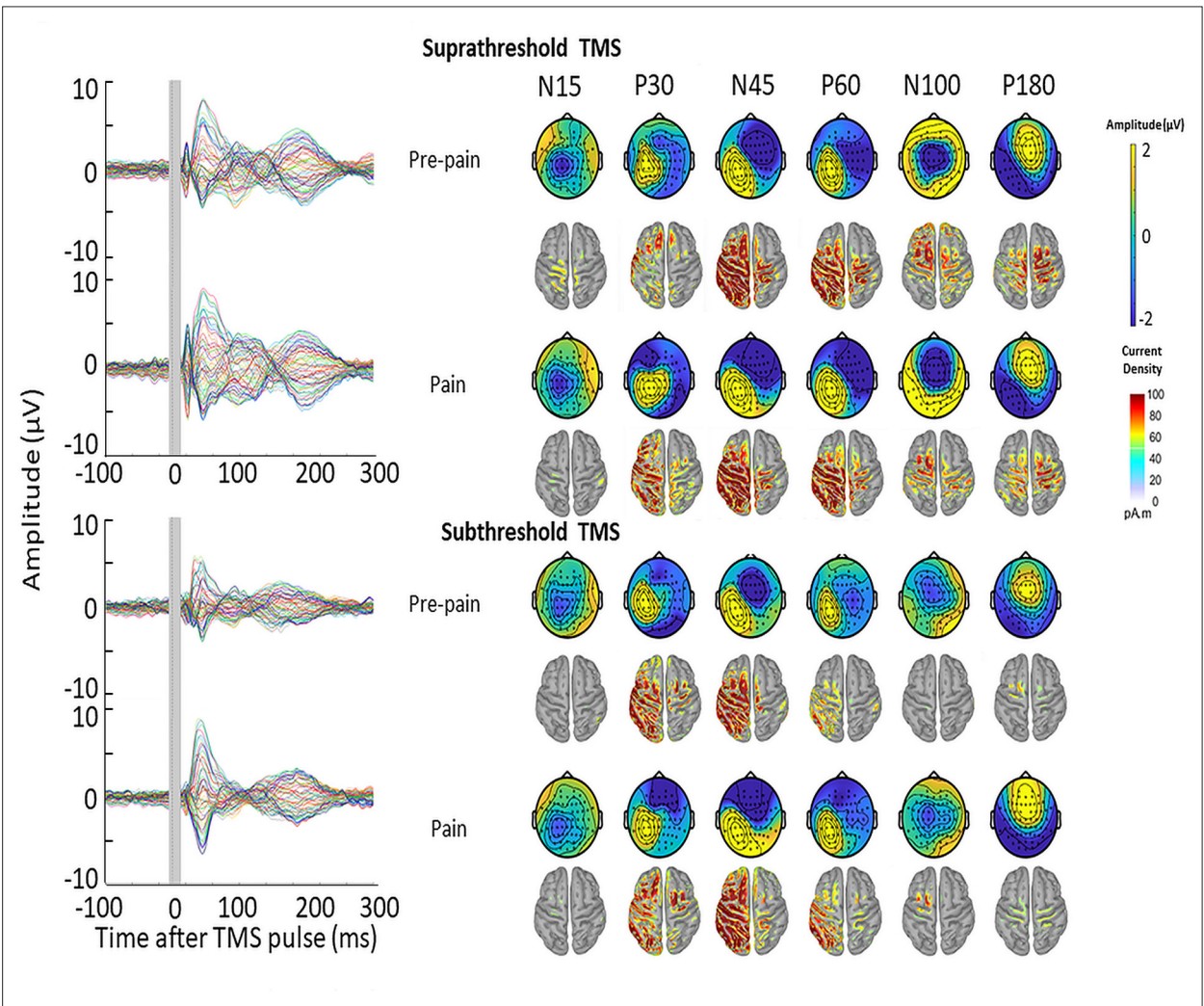

**Figure 9.** TMS-evoked potentials for supra- and subthreshold TMS. Left: TEPs (n = 10) during the pre-pain and pain blocks, for both supra- and subthreshold TMS of Experiment 3. The gray-shaded area represents the window of interpolation around the transcranial magnetic stimulation TMS pulse. Right: Scalp topographies and estimated source activity at timepoints where TEP peaks are commonly observed, including the N15, P30, N45, P60, N100, and P180.

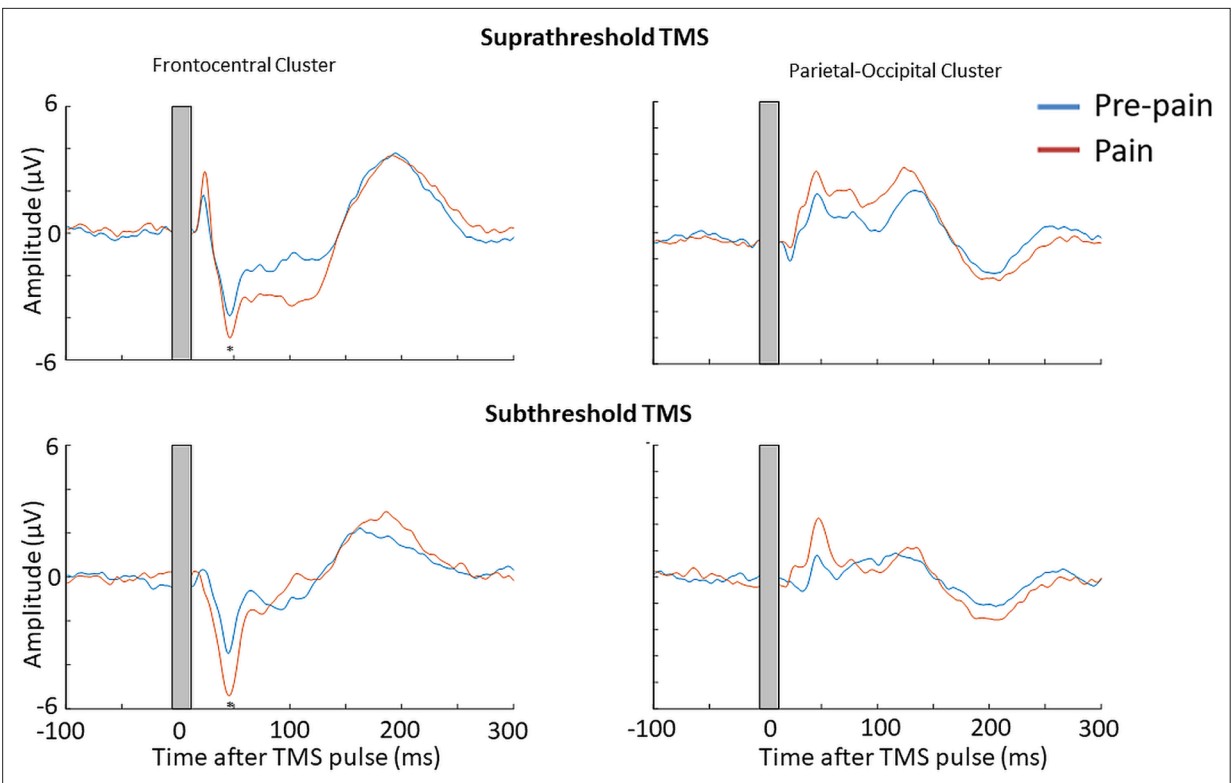

**Figure 10.** Pain led to an increase in the N45 peak amplitude for both suprathreshold and subthreshold TMS. TEPs (n = 10) during pain and pre-pain blocks of Experiment 3, across supra- and subthreshold TMS conditions, for the frontocentral electrodes (left) and parietal occipital electrodes (right) identified from the cluster analysis in Experiment 1. A significantly stronger frontocentral negative peak was identified ~45ms post-TMS during pain compared to pre-pain for both supra- and subthreshold stimulation. The astericks indicates at least moderate Bayesian evidence for the alternative hypothesis that the amplitude is larger in pain vs. pre-pain (BF$_{10}$ >3).

## Pain ratings

The mean cold and warm detection threshold was 29.4 ± 2.0°C and 34.9 ± 1.4°C, respectively. The mean heat pain threshold was 42.9 ± 2.5°C. *Figure 3—figure supplement 2* and *Figure 3—figure supplement 3* shows, respectively, the warmth ratings during the pre-pain block for each condition, and the pain ratings during the pain block for each condition.

## TMS-evoked potentials

The mean RMT, subthreshold and suprathreshold stimulus intensity was, respectively, 68.0 ± 8.3%, 61.0 ± 7.3% and 74.8 ± 9.1% of maximum stimulus output. *Figure 9* shows the grand average TEPs, scalp topographies and estimated source activity for supra- and subthreshold TMS, across pre-pain and pain blocks. *Figure 10* shows the mean TEP waveform for the frontocentral and parietal-occipital clusters identified from Experiment 1, across suprathreshold and subthreshold TMS conditions. When comparing the pain with pre-pain blocks, there was moderate evidence that the frontocentral N45 was increased during subthreshold TMS (BF$_{10}$=3.05) and suprathreshold TMS (BF$_{10}$=3.01). When comparing pain with pre-pain blocks, there was anecdotal evidence for no alterations in the frontocentral N100 peak during suprathreshold TMS (BF$_{10}$=0.42) and subthreshold TMS (BF$_{10}$=0.36). When comparing pain with pre-pain blocks, there was anecdotal evidence for an increase in the parietal occipital P60 peak during suprathreshold TMS (BF$_{10}$=2.71) and anecdotal evidence for no alteration in the P60 peak during subthreshold TMS (BF$_{10}$=0.72). Overall, there was evidence that that the N45 peak was altered in response to both supra- and subthreshold TMS, suggesting the pain-induced increase in the N45 peak was not a result of stronger reafferent feedback from the muscle twitches.

## Stability of N45 peaks across experiments

We conducted a supplementary investigation of the stability of the N45 TEP peaks for each experiment. The interclass correlation coefficient (Two-way fixed, single measure) for the N45 to active suprathreshold TMS across timepoints for each experiment was 0.90 for Experiment 1 (across pre-pain, pain, post-pain time points), 0.74 for Experiment 2 (across pre-pain and pain conditions), and 0.95 for Experiment 3 (across pre-pain and pain conditions). This suggests that even with the fluctuations in the N45 induced by pain, the N45 for each participant was stable across time, further supporting the reliability of our data. Data supporting the findings of this study are available on Open science framework https://osf.io/k3psu/.

## Discussion

The present study determined whether acute experimental pain induces alterations in cortical inhibitory and/or facilitatory activity observed in TMS-evoked potentials. Across three experiments, there was Bayesian evidence (varying between moderate to very strong) for an increase in the amplitude of the N45 peak during painful stimuli compared to a non-painful baseline. Experiment 1 showed very strong evidence that a larger increase in the N45 peak in response to pain was correlated with higher pain ratings. Experiment 2 showed that the increase in the N45 peak during pain was not a result of alterations in sensory potentials associated with the TMS pulses, but rather, changes in cortical excitability. Experiment 3 showed that the increase in the N45 peak was not a result of stronger reafferent feedback from muscle twitches evoked by TMS during painful stimuli. While Experiment 1 showed moderate evidence for an increase in N100 and P60 peaks during pain relative to pre-pain baseline, this was not replicated in the follow-up experiments. Experiment 1 showed anecdotal evidence for group-level alterations in MEP amplitude during pain; however, there was very strong evidence that a larger reduction in MEP amplitude during pain was correlated with lower pain ratings.

### Increased GABAergic activity during tonic pain

This study is the first to use TMS-EEG methodology to examine the direct cortical response to acute pain, extending previous studies that have used TMS to measure MEPs in response to pain (*Chowdhury et al., 2022a*). The key finding was an increase in the amplitude of the N45 peak in response to pain. This result was replicated across three experiments, providing robust evidence for the effect. Furthermore, we accounted for major confounds that have caused significant data interpretation issues in the TMS-EEG literature in recent years, namely the contamination of TEPs by sensory potentials associated with TMS pulses (*Biabani et al., 2019*; *Chowdhury et al., 2022b*) and the presence of reafferent feedback from muscle twitches (*Ahn and Fröhlich, 2021*).

The finding of a reliable increase in the amplitude of the N45 peak during pain suggests a role for $GABA_A$ neurotransmission in pain processing, as previous work has shown that the amplitude of the N45 peak is increased in response to $GABA_A$ agonists (*Premoli et al., 2014*). Our source reconstruction results suggest that around this timepoint, the current density was stronger in the sensorimotor area, consistent with the idea that the N45 peak reflects GABAergic activity within the sensorimotor cortex (*Farzan and Bortoletto, 2022*). While it has been shown that reafferent muscle activity also contributes to the N45 peak (*Ahn and Fröhlich, 2021*), Experiment 3 showed that pain increased the amplitude of the N45 peak even during subthreshold TMS. Taken together, these findings suggest that the increased amplitude of the N45 peak in response to pain reflects an increase in GABAergic activity within the sensorimotor cortex.

GABAergic neurons play a critical role in pain-related brain networks (*Barr et al., 2013*; *Ong et al., 2019*). They are involved in the generation of gamma oscillations (*Buzsáki and Wang, 2012*), which have been strongly implicated in pain perception (*Barr et al., 2013*; *Li et al., 2023*). Indeed, previous work has shown an increase in gamma oscillations in response to painful thermal stimuli comparable to the present study, across a wide range of brain regions such as the prefrontal (*Schulz et al., 2015*) and sensorimotor cortices (*Gross et al., 2007*). It is therefore possible that increases in the N45 peak during pain reflect increased sensorimotor gamma oscillations. Further multimodal work is required to confirm this finding.

While our findings are consistent with some studies that show increases in GABAergic activity in response to pain (*Gross et al., 2007*; *Kupers et al., 2009*; *Schulz et al., 2015*), other studies have

also reported reduced GABAergic activity in response to experimental pain (*Cleve et al., 2015*; *de Matos et al., 2017*). Differences between studies can be attributed to the duration of the noxious stimulus (tonic pain lasting several seconds/minutes vs. transient pain stimuli lasting <1 s). Indeed, pooled data have shown that the cerebral response to pain is highly dependent on the duration of the painful stimuli, as the adaptive response (to suppress or increase cortical activity) changes depending on the duration of pain (*Chowdhury et al., 2022a*). This highlights the need for further work to replicate our findings using different durations of experimental pain and in chronic pain populations.

Another finding of Experiment 1 was the increase in the amplitude of N100 peak, a marker of GABA$_B$ neurotransmission (*Premoli et al., 2014*), and the parietal-occipital P60 peak, a marker of glutamatergic neurotransmission (*Belardinelli et al., 2021*). However, this was not replicated in Experiments 2 and 3, potentially due to the smaller sample size. Nonetheless, we encourage further investigations of alterations in these peaks during pain, particularly the P60 peak, as several magnetic resonance spectroscopy studies have reported increases in glutamate concentration during experimental pain (*Archibald et al., 2020*).

## Predicting individual differences in pain using TEPs

Experimental pain models are useful tools to explore brain measures that may predict individual differences in pain sensitivity, with an ultimate goal of determining whether such measures explain why some people develop chronic pain. Experiment 1 showed that higher pain ratings were associated with a larger increase in the N45 peak during pain. This analysis was not conducted in Experiments 2 and 3 due to smaller sample sizes and given the primary aims of Experiments 2 and 3 were to isolate the source of the group-level effect. Nonetheless, our results suggest that the N45 peak is a potential marker of sensorimotor GABAergic activity and may be associated with individual differences in pain sensitivity. This is consistent with other studies measuring GABAergic responses to pain, showing associations between higher pain sensitivity and larger sensorimotor gamma oscillations (*Barr et al., 2013*) and higher left somatosensory cortical GABA laterality (*Niddam et al., 2021*). However, the direction of this relationship likely depends on the duration of pain (*Chowdhury et al., 2022a*). Our results have implications for understanding the development and maintenance of chronic pain. Further TMS-EEG studies are required to determine whether the N45 peak is altered in chronic pain populations and whether the N45 peak can explain why some individuals in the acute stages of pain transition to chronic pain.

## The TEP vs. MEP response to pain

The present study showed that a larger reduction in MEP amplitude during pain was correlated with lower pain ratings, consistent with a recent systematic review (*Chowdhury et al., 2022a*) and the idea that reduced MEP amplitude is an adaptive mechanism that restricts movement in the pain-afflicted area, to protect the area from further pain and injury (*Hodges and Tucker, 2011*). The novelty of this study was the use of an experimental heat pain paradigm that has not yet been used in combination with TMS research, and a paradigm that controls for non-painful somatosensory stimulation.

The finding of a pain-induced increase in the amplitude of the N45 peak, which indexes GABA$_A$ receptor activity, is consistent with TMS research showing pain-induced increases in short-interval intracortical inhibition (SICI) (*Salo et al., 2019*; *Schabrun and Hodges, 2012*). SICI refers to the reduction in MEP amplitude to a TMS pulse that is preceded 1–5ms by a subthreshold pulse, with this reduction believed to be mediated by GABA$_A$ neurotransmission (*Kujirai et al., 1993*). Some studies have reported associations between SICI and the TEP N45 peak (*Leodori et al., 2019*; *Rawji et al., 2019*), suggesting the two may share common neurophysiological mechanisms. However, we also found that a larger reduction in MEP amplitude during pain was associated with less pain, while a larger increase in the TEP N45 peak during pain was associated with stronger pain ratings, suggesting that inhibitory processes mediating MEPs and the TEP N45 peak during pain are distinct. Further work is required to disentangle the relationship between corticomotor excitability measured by MEPs and cortical activity measured by TEPs.

## Study limitations

Some methodological limitations should be noted. Firstly, while there was no conclusive evidence for a difference in pain ratings between the six thermal stimuli of the pain block, there was evidence

for fluctuations in pain ratings during each painful stimulus. This suggests that the perceived pain intensity was not stable across 40 s, which may have introduced noise in the TEP data. Future studies could use pain paradigms that can more effectively maintain a constant level of pain, for example hypertonic saline infusion paradigms (*Svensson et al., 2003*). Secondly, the use of verbal pain ratings prevented the characterization of pain on a finer time scale. However, verbal ratings were used to eliminate potential contamination of MEPs introduced by using the hand for providing pain rating. Thirdly, the increased N45 peak amplitude in response to pain may reflect increased alertness/arousal during pain. However, a recent study showed that higher alertness is associated with reduced TEP amplitude (*Noreika et al., 2020*), suggesting the increase in the N45 peak amplitude is not related to pain-induced arousal. Lastly, future research should consider replicating our experiment using inter-mixed pain and no pain blocks, as opposed to fixed pre-pain and pain blocks, to control for order effects (the explanation that successive thermal stimuli applied to the skin results in an increase in the N45 peak, regardless of whether the stimuli are painful or not). However, we note that there was no conclusive evidence for a difference in N45 peak amplitude between pre-pain and post-pain conditions of Experiment 1 (*Figure 6—figure supplement 1*), suggesting it is unlikely that the observed effects were an artefact of time.

## Conclusion

This study is the first to use TMS-EEG methodology to examine alterations in cortical activity in direct response to acute pain. Findings across three experiments suggest that tonic heat pain leads to an increase in the amplitude of the frontocentral TEP N45 peak (associated with GABAergic neurotransmission), and that larger increases in this peak are associated with higher pain ratings. The findings suggest that TEP indices of GABAergic neurotransmission have the potential to be used as predictive markers of pain severity.

## Materials and methods

### Participants

Experiment 1 consisted of 29 healthy participants (18 males, 11 females, mean age; 26.24±5.5). Participants were excluded if they had a history of chronic pain condition or any current acute pain, any contraindication to TMS such as pregnancy, mental implants in the skull, seizure, or if they reported a history of neurological or psychiatric conditions, or were taking psychoactive medication. Participants completed a TMS safety screen (*Rossi et al., 2009*). Procedures adhered to the Declaration of Helsinki and were approved by the human research ethics committee of UNSW (HC200328). All participants provided informed written consent.

The sample size calculation was done in G*power 3.1.9.7 with 80% power. As there were no prior TMS-EEG pain studies, we used pooled data from our systematic review on TMS studies (*Chowdhury et al., 2022a*) showing that the weighted effect size of changes in MEP amplitude in response to tonic experimental pain was 0.56. Using this value, 28 participants were required to detect a significant difference between pain and pre-pain blocks.

To determine the sample size of Experiments 2 and 3, we computed the effect sizes of the N45, P60 and N100 changes (pain vs. prepain) from Experiment 1 (Cohen's $d_{RM}$ = 1.76, 0.99, and 0.83 for N45, P60, and N100, respectively). With a power of 80%, the required sample size was 4–11 participants to detect a significant difference. Experiment 2 recruited a further 10 healthy participants (four males, six females, mean age: 26.8±5.9) and Experiment 3 consisted of 10 healthy participants (four males, six females, mean age: 28±5.9).

### Experimental protocol

Participants were seated comfortably in a shielded room. They viewed a fixation cross to minimize eye movements. TMS was applied to left M1 while participants wore an EEG cap containing 63 scalp electrodes to record TEPs. Surface electromyographic (EMG) electrodes were placed over the distal region of the right ECRB to record MEPs. EMG signals were amplified (x 1000) and filtered (16–1000 Hz), and digitally sampled at 2000 Hz (Spike2, CED). A thermode was attached over the proximal region of the right ECRB in close proximity to the EMG electrodes (*Figure 1*). The TMS coil was covered in a layer of foam (5 mm thickness) to minimize decay artefacts (*Rogasch et al., 2017*). Participants also wore

both foam earplugs and headphones to reduce any potential discomfort from the TMS click. Auditory masking was not used. Instead, auditory-evoked potentials resulting from the TMS click sound were controlled for in Experiment 2.

The protocol for Experiment 1 is illustrated in *Figure 2* (*Furman et al., 2020*; *Granot et al., 2006*). In Experiment 1, participants experienced three blocks of thermal stimuli, in chronological order: pre-pain, pain, and post-pain block. Each block consisted of multiple thermal stimuli delivered 40 s at a time during which suprathreshold (110% RMT) TMS measurements and verbal pain ratings were obtained. The thermode commenced at a baseline temperature of 32 °C. The pre-pain and post-pain blocks consisted of six thermal stimuli delivered at the warm threshold (the temperature that led to any detectable change in skin temperature from baseline). In the pain block, six thermal stimuli were delivered at 46 °C, which has been shown to produce lasting pain with a mean rating of ~5/10 (*Furman et al., 2020*). Given we were interested in the individual relationship between pain and excitability changes, the fixed temperature of 46 °C ensured larger variability in pain ratings as opposed to calibrating the temperature of the thermode for each participant (*Adamczyk et al., 2022*). The inclusion of blocks with warm stimuli allowed for control for changes in cortical excitability due to non-painful somatosensory stimulation.

The protocol for Experiment 2 (*Figure 2—figure supplement 1*) and Experiment 3 (*Figure 2—figure supplement 2*) were identical to Experiment 1 with two differences: the exclusion of the post-pain block (as the aim was to disentangle the source of the pain vs. pre-pain effect from Experiment 1) and the inclusion of a sham TMS condition (Experiment 2) or subthreshold (90% RMT) TMS (Experiment 3) intermixed within both the pre-pain and pain blocks.

## Electrical stimulation setup (Experiment 2 Only)

Electrical stimulation was based on previous studies attempting to simulate the somatosensory component of active TMS (*Chowdhury et al., 2022b*; *Gordon et al., 2021*; *Rocchi et al., 2021*). Prior to EEG setup, 8 mm Ag/AgCl electrodes were placed directly over the scalp. 'Snap on' lead wires were then clipped in place and connected to the electrical stimulator (Digitimer DS7AH, Digitimer Ltd., UK). To keep the electrodes and lead wires firmly in position, participants were fitted with a tight netted wig cap, which sat on top of the electrodes but underneath the EEG cap. Consistent with previous research (*Chowdhury et al., 2022b*; *Rocchi et al., 2021*), and to minimize EEG artefacts caused by electrical stimulation, the stimulating electrodes were not placed directly underneath the EEG electrodes. Rather, stimulating electrodes were positioned in the middle of the EEG electrode cluster located in closest proximity to the motor hotspot. This roughly corresponded to an anode position between FC1 and FC3 and a cathode position between C1 and C3. Scalp electrical stimulation was delivered using a 200 µs square wave via with a compliance of 200 V.

## Electroencephalography

EEG was recorded using a DC-coupled, TMS-compatible amplifier (ActiChamp Plus, Brain Products, Germany) at a sampling rate of 25,000 Hz. Signals were recorded from 63 TMS-compatible active electrodes (6 mm height, 13 mm width), embedded in an elastic cap (ActiCap, Brain Products, Germany), in line with the international 10–10 system. Active electrodes result in similar TEPs (both magnitude and peaks) to more commonly used passive electrodes (*Mancuso et al., 2021*). There is also evidence that active electrodes have higher signal quality than passive electrodes at higher impedance levels (*Laszlo et al., 2014*). Recordings were referenced online to 'FCz' and the ground electrode placed on 'FPz'. Electrolyte gel was used to reduce electrode impedances below ~5kOhms. Online TEP monitoring was not available with the EEG software.

## Transcranial magnetic stimulation

Single, monophasic stimuli were delivered using a Magstim unit (Magstim Ltd., UK) and 70 mm figure-of-eight flat coil. The coil was oriented at 45° to the midline, inducing a current in the posterior-anterior direction. The scalp site that evoked the largest MEP measured at the ECRB ('hotspot') was determined and marked. The RMT was determined using the TMS motor thresholding assessment tool, which estimates the TMS intensity required to induce an MEP of 50 microvolts with a 50% probability using maximum likelihood parametric estimation by sequential testing (*Awiszus, 2003*; *Awiszus and Borckardt, 2011*). This method has been shown to achieve the accuracy of methods such as the

Rossini-Rothwell method (*Rossini et al., 1994*; *Rothwell et al., 1999*) but with fewer pulses (*Qi et al., 2011*; *Silbert et al., 2013*). The test stimulus intensity was set at 110% RMT to concurrently measure MEPs and TEPs during pre-pain, pain and post-pain blocks.

### Thermal pain

Thermal stimuli were delivered over the proximal region of the right ECRB using a contact heat stimulator (27 mm diameter Medoc Pathway CHEPS Peltier device; Medoc Advanced Medical Systems Ltd). Pain ratings were obtained after each TMS pulse using a verbal rating scale (0=no pain, and 10=most pain imaginable). Verbal ratings were collected rather than pain ratings provided on the computer by hand to avoid contamination of MEP measures from motor processes of hand movements. Verbal pain ratings have been shown to yield excellent test-retest reliability (*Alghadir et al., 2018*).

### Quantitative sensory testing

Warmth, cold and pain thresholds were assessed in line with a previous study (*Furman et al., 2020*). With the baseline temperature set at a neutral skin temperature of 32 °C, participants completed three threshold tests: to report when they felt a temperature increase (warmth detection threshold; *Furman et al., 2020*), to report when they felt a temperature decrease (cool detection threshold; *Furman et al., 2020*); (3) to report when an increasing temperature first became painful (heat pain threshold; *Furman et al., 2020*). A total of three trials was conducted for each test to obtain an average, with an interstimulus interval of 6 s (*Furman et al., 2020*). The sequence of cold, warmth and pain threshold was the same for all participants. Participants provided feedback for each trial by pressing a button (with their left hand) on a hand-held device connected to the Medoc Pathway. Temperatures were applied with a rise/decrease rate of 1 °C/s and return rate of 2 °C/s (initiated by the button click).

### Matching task (Experiment 2 only)

As the aim of Experiment 2 was to perceptually match the somatosensory aspects of active and sham TMS, a 2-Alternative Forced Choice task was used to determine the electrical stimulation intensity that led to a similar flicking sensation to active TMS (*Chowdhury et al., 2022b*). Participants received either electrical stimulation or active TMS in a randomized order and were asked whether the first or second stimulus led to a stronger flick sensation. The electrical stimulation intensity was then increased or decreased until participants could no longer judge the first or second stimulus as stronger. This intensity was then applied during the test blocks.

### Test blocks
#### Experiment 1

The temperature of the thermode commenced at a neutral skin temperature (32 °C). Participants were exposed to 18 sustained thermal stimuli with a 20-s interstimulus interval. For each thermal stimulus, a single temperature (rise rate of 1 °C/s, return rate of 2 °C/s) was applied over the proximal region of the ECRB for 40 s. Thermal stimuli 1–6 (pre-pain block) were delivered at the participant's individually determined warmth detection threshold, Thermal stimuli 7–12 at 46°, and Thermal stimuli 13–18 again at the participant's warmth detection threshold. Participants were not informed of the order of the warm and painful stimuli to minimize the influence of expectation of pain on TEPs and MEPs. During each 40-s thermal stimulus, TMS pulses were manually delivered, with a verbal pain rating score (0=no pain, and 10=worst pain imaginable) obtained between pulses. To avoid contamination of TEPs by verbal ratings, the subsequent TMS pulse was not delivered until the verbal rating was complete, and the participant was cued by the experimenter to provide the pain rating after each pulse. As TMS was delivered manually, there was no set interpulse interval. However, the 40-s thermal stimulus duration allowed for 11 pulses for each thermal stimulus (hence 66 TMS pulses for each of the pre-pain, pain, and post-pain blocks), and 10 verbal pain ratings between each TMS pulse (~4 s in between pain ratings). Current recommendations (*Hernandez-Pavon et al., 2023*) suggest basing the number of TMS trials per condition on the key outcome measure (e.g. TEP peaks vs. frequency measures) and based on previous test-retest reliability studies. In our study the number of trials was based on a test-retest reliability study by (*Kerwin et al., 2018*) which showed that 60 TMS pulses

(delivered in the same run) was sufficient to obtain reliable TEP peaks (i.e. sufficient within-individual concordance between the resultant TEP peaks of each trial).

### Experiment 2

Participants were exposed to 24 sustained thermal stimuli (40 s each). Thermal stimuli 1–6 and 7–12 consisted of warm stimuli (pre-pain block), while Thermal stimuli 13–18 and 19–24 consisted of stimuli delivered at 46 °C (pain block). Active or sham TMS was delivered during either thermal stimuli 1–6 or 7–12, with the order of active and sham randomly determined for each participant. The same applied for thermal stimuli 13–18 and 19–24. The active and sham TMS conditions were similar to that used in a recent TMS-EEG study (*Gordon et al., 2021*). Sham TMS involved the active TMS coil rotated 90° to the scalp, and a sham coil (identical in shape/weight) placed underneath the active coil and tangentially over the scalp. The active TMS coil was then triggered with the electrical stimulation unit to simultaneously simulate the auditory and somatosensory components of active TMS, respectively. Active TMS involved the delivery of the active TMS coil placed tangentially over the scalp, and the sham TMS coil above the active coil rotated 90° to the scalp (see *Figure 7*). The design allowed for 11 pulses for each thermal stimulus and ten pain ratings (hence 66 TMS pulses for active pre-pain, sham pre-pain, active pain and sham pain blocks).

### Experiment 3

Participants were exposed to 24 sustained thermal stimuli (40 s each). Thermal stimuli 1–6 and 7–12 consisted of warm stimuli (pre-pain block), while thermal stimuli 13–18 and 19–24 consisted of thermal stimuli delivered at 46 °C (pain block). Suprathreshold or subthreshold TMS (90% RMT) was delivered during either thermal stimuli 1–6 or 7–12, with the order of supra- and subthreshold TMS randomly determined for each participant. The same applied for thermal stimuli 13–18 and 19–24. In addition to the pain rating in between TMS pulses, we collected a second rating for warmth of the thermal stimulus (0=neutral, 10=very warm) to confirm that the participants felt some difference in sensation relative to baseline during the pre-pain block. Overall, the design allowed for 11 pulses for each thermal stimulus and 10 pain/warmth ratings (hence 66 TMS pulses for suprathreshold pre-pain, subthreshold pre-pain, suprathreshold pain and subthreshold pain blocks).

## Data processing

### Motor-evoked potentials

The amplitude of each MEP was determined using a custom MATLAB script (https://github.com/Nahian92/TMS_EEG_Preprocessing/tree/main, copy archived at *Chowdhury, 2023*). The onsets and offsets of the MEPs were manually determined for each trial. In some participants, background EMG activity was observed due to placement of the thermode close to the EMG electrodes, which can influence MEP amplitude (*Ruddy et al., 2018*). To account for this, MEP amplitude was calculated by subtracting the root mean square (RMS) of background EMG noise from the RMS of the MEP window using a fixed window between 55 and 5ms before the TMS pulse (*Chowdhury et al., 2023*; *Tsao et al., 2011*).

### TMS-evoked potentials

Pre-processing of the TEPs was completed using EEGLAB (*Delorme and Makeig, 2004*) and TESA (*Rogasch et al., 2017*) in MATLAB (R2021b, The Math works, USA), and based on previously described methods (*Chowdhury et al., 2022b*; *Mutanen et al., 2018*; *Rogasch et al., 2017*). The script is available on available on https://github.com/Nahian92/TMS_EEG_Preprocessing/tree/main, (copy archived at *Chowdhury, 2023*). First, bad channels were removed. The mean number of channels removed across participants was 2.2±2.7 for Experiment 1, 3±2.2 for Experiment 2, and 6.1±2.64 for Experiment 3. The period between –5 and ~14ms after the TMS pulse was removed and interpolated using the ARFIT function for continuous data (*Neumaier and Schneider, 2001*; *Schneider and Neumaier, 2001*). The exact interval was based on the duration of decay artefacts. Data was epoched 1000ms before and after the TMS pulse, and baseline corrected between –1000 and –5ms before the TMS pulse. Noisy epochs were identified via the EEGLAB auto-trial rejection function (*Delorme et al., 2007*) and then visually confirmed. There was moderate Bayesian evidence for no

difference in the mean number of epochs excluded between conditions for Experiment 1 (5.8±5.2, 5.1±5.0 and 5.3±4.1 for the pre-pain, pain, and post-pain conditions, respectively, $BF_{10}$=0.145), Experiment 2 (5.1±3.0, 5.6±3.4, 8.2±5.2 and 4.6±4.4 for the active pre-pain, sham pre-pain, active pain, and sham pain conditions, respectively, $BF_{10}$=0.27) and Experiment 3 (5.4±3.4, 8.1±6.5, 6.5±6.6 and 5.4±2.8 for the subthreshold pre-pain, suprathreshold pre-pain, subthreshold pain, suprathreshold pain, respectively, $BF_{10}$=0.169). The fastICA algorithm with auto-component rejection was used to remove eyeblink and muscle artefacts (*Rogasch et al., 2017*). There was anecdotal or moderate Bayesian evidence for no difference in the mean number of rejected components between conditions for Experiment 1 (11.0±6.3, 11.7±8.4 and 10.25±8.3 for the pre-pain, pain, and post-pain block, respectively, $BF_{10}$=0.19), Experiment 2 (11.4±8.4, 12.3±6.9, 10.1±3.4 and 12.9±9.5 for the active pre-pain, sham pre-pain, active pain, and sham pain conditions, respectively, $BF_{10}$=0.181) and Experiment 3 (9.1±7.8, 7.6±7.1, 9.6±7.8 and 8.6±7.0, $BF_{10}$=0.576). The source-estimation noise-discarding (SOUND) algorithm was applied (*Mutanen et al., 2020*; *Mutanen et al., 2018*), which estimates and supresses noise at each channel based on the most likely cortical current distribution given the recording of other channels. This signal was then re-referenced (to average). A band-pass (1–100 Hz) and band-stop (48–52 Hz) Butterworth filter was then applied. Any lost channels were interpolated.

## Source localization

Source localization of TEPs was conducted using Brainstorm (*Tadel et al., 2011*). A template brain model (ICBM 152) was co-registered with the TMS-EEG data. Noise estimation was used to determine sensor weighting and regularization parameter of the current density construction. The forward model involved use of the Symmetric Boundary Element Method with the head having 3 compartments of fixed conductivities, implemented in OpenMEEG software (*Gramfort et al., 2010*), and inverse model involved use of Minimum Norm Estimations.

## Statistical analysis

Given we were interested in determining the evidence for pain altering TEP peaks in certain conditions (e.g. active TMS) and pain not altering TEP peaks in other conditions (sham TMS), we used a Bayesian approach as opposed to a frequentist approach, which considers the strength of the evidence for the alternative vs. null hypothesis. Bayesian inference was used to analyze the data using JASP software (Version 0.12.2.0, JASP Team, 2020). Bayes factors were expressed as $BF_{10}$ values, where $BF_{10}$'s of 1–3, 3–10, 10–30 and 30–100 indicated 'weak', 'moderate', 'strong' and 'very strong' evidence for the alternative hypothesis, while $BF_{10}$'s of 1/3–1, 1/10-1/3, 1/30-1/10 and 1/100-1/30 indicated 'anecdotal', 'moderate', 'strong' and 'very strong' evidence in favour of the null hypothesis (*van Doorn et al., 2021*).

### Pain ratings

A 6 (thermal stimulus number: 1–6) x 10 (timepoint:1–10) Bayesian repeated measures ANOVA with default priors in JASP (r scale fixed effects = .5, r scale random effects = 1, r scale covariates = .354) was conducted on the pain ratings during the pain block. This was to assess differences in pain ratings between the six painful stimuli and whether pain ratings differed between the timepoints of each painful stimulus.

### Motor-evoked potentials

A Bayesian one-way repeated measures ANOVA with default priors in JASP was performed to assess differences in MEP amplitudes between pre-pain, pain, and post-pain blocks of Experiment 1. As there is now increasing emphasis placed on investigating the individual level relationship between changes in cortical excitability and pain and not only the group level effect (*Chowdhury et al., 2022a*; *Seminowicz et al., 2018*; *Seminowicz et al., 2019*; *Summers et al., 2019*), we also investigated the correlations between pain ratings and changes in MEP (and TEP) amplitude, A Bayesian correlation analysis with default priors in JASP (stretched Beta prior width = 1) was run to determine whether the change in mean MEP amplitude during pain (as a proportion of pre-pain) was associated with the mean verbal pain rating score. Data were checked for assumptions of normally distributed data using a Shapiro-Wilk test. Where assumptions were violated, data were log-transformed.

## TMS-evoked potentials

The grand-averaged signals for the pre-pain, pain, and post-pain condition were obtained. For Experiment 1, a cluster-based permutation analysis was used to compare amplitude levels between pre-pain and pain, and pre-pain and post-pain, at each time-point and for each electrode. For all experiments, the mean TEP waveform of any identified clusters from Experiment 1 were plotted, and peaks (e.g. N15, P30, N45, P60, N100) were identified using the TESA peak function (*Rogasch et al., 2017*). Any identified peaks were then compared between conditions using Bayes paired sample t-tests with default priors in JASP (Cauchy scale = .707). A Bayesian correlation analysis with default priors in JASP was performed to determine whether the difference in identified peaks between pre-pain and pain blocks was associated with the mean pain rating score. Data were checked for assumptions of normally distributed data using a Shapiro-Wilk test. Where assumptions were violated, data were log-transformed.

## Acknowledgements

This work was supported by 1R61NS113269-01 from The National Institutes of Health to DAS, SMS, and The Four Borders Foundation to DAS.

## Additional information

### Funding

| Funder | Grant reference number | Author |
| --- | --- | --- |
| National Institutes of Health | 1R61NS113269-01 | David A Seminowicz Siobhan M Schabrun |

The funders had no role in study design, data collection and interpretation, or the decision to submit the work for publication.

### Author contributions

Nahian Shahmat Chowdhury, Conceptualization, Data curation, Software, Formal analysis, Investigation, Methodology, Writing – original draft, Project administration, Writing – review and editing; Alan KI Chiang, Conceptualization, Investigation, Methodology, Project administration, Writing – review and editing; Samantha K Millard, Conceptualization, Investigation, Methodology, Writing – review and editing; Patrick Skippen, Formal analysis, Writing – review and editing; Wei-Ju Chang, Conceptualization, Writing – review and editing; David A Seminowicz, Siobhan M Schabrun, Conceptualization, Resources, Supervision, Writing – review and editing

### Author ORCIDs

Nahian Shahmat Chowdhury ⓘ https://orcid.org/0000-0001-6357-0746
Alan KI Chiang ⓘ https://orcid.org/0000-0002-9156-0534
Samantha K Millard ⓘ https://orcid.org/0000-0003-1409-8179
Wei-Ju Chang ⓘ https://orcid.org/0000-0003-0524-4883
David A Seminowicz ⓘ https://orcid.org/0000-0003-3111-3756

### Ethics

Procedures adhered to the Declaration of Helsinki and were approved by the human research ethics committee of UNSW (HC200328). All participants provided informed written consent.

Reviewer #1 (Public Review): https://doi.org/10.7554/eLife.88567.3.sa1
Reviewer #2 (Public Review): https://doi.org/10.7554/eLife.88567.3.sa2
Reviewer #3 (Public Review): https://doi.org/10.7554/eLife.88567.3.sa3
Author Response https://doi.org/10.7554/eLife.88567.3.sa4

## Additional files

### Supplementary files
• MDAR checklist

### Data availability
Data supporting the findings of this study are available on Open science framework https://osf.io/k3psu/.

The following dataset was generated:

| Author(s) | Year | Dataset title | Dataset URL | Database and Identifier |
|---|---|---|---|---|
| Chowdhury N | 2023 | Alterations in cortical excitability during pain: A combined TMS-EEG Study | https://osf.io/k3psu/ | Open Science Framework, k3psu |

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
