## [Editor Report · eLife assessment]

This **valuable** study provides **convincing** evidence that acute experimental pain induces changes of cortical excitability. Although the modality specificity of the findings is not fully clear, the findings will be of interest to researchers interested in the brain mechanisms of pain.

---

## [Referee Report · Reviewer #1 (Public Review)]

The objective of this investigation was to determine whether experimental pain could induce alterations in cortical inhibitory / facilitatory activity observed in TMS-evoked potentials (TEPs). Previous TMS investigations of pain perception had focused on motor evoked potentials (MEPs), which reflect a combination of cortical, spinal, and peripheral activity, as well as restricting the focus to M1. The main strength of this investigation is the combined use of TMS and EEG in the context of experimental pain. More specifically, Experiment 1 investigated whether acute pain altered cortical excitability, reflected in the modulation of TEPs. The main outcome of this study is that relative to non-painful warm stimuli, painful thermal stimuli led to an increase on the amplitude of the TEP N45, with a larger increase associated with higher pain ratings. Because it has been argued that a significant portion of TEPs could reflect auditory potentials elicited by the sound (click) of the TMS, Experiment 2 constituted a control study that aimed to disentangle the cortical response related to TMS and auditory activity. Finally, Experiment 3 aimed to disentangle the cortical response to TMS and reafferent feedback from muscular activity elicited by suprathreshold TMS applied over M1. The fact that the authors accompanied their main experiment with two control experiments strengthens the conclusion that the N45 TEP peak could be implicated in the perception of painful stimuli. Perhaps, the addition of a highly salient but non-painful stimulus (i.e. from another modality) would have further ruled out that the effects on the N45 are not predominantly related to intensity / saliency of the stimulus rather than to pain per se.

---

## [Referee Report · Reviewer #2 (Public Review)]

The authors have used transcranial magnetic stimulation (TMS) and motor evoked potentials (MEPs) and TMS-electroencephalography (EEG) evoked potentials (TEPs) to determine how experimental heat pain could induce alterations these metrics.In Experiment 1 thermal stimuli were administered over the forearm, with the first, second and third block of stimuli consisting of warm but non painful (pre-pain block), painful heat (pain block) and warm but non-painful (post-pain block) temperatures respectively. Painful stimuli led to an increase in the amplitude of the fronto-central N45, with a larger increase associated with higher pain ratings. Experiments 2 and 3 studied the correlation between the increase in the N45 in pain and the effects of a sham stimulation protocol/higher stimulation intensity. They found that the centro-frontal N45 TEP was decreased in acute pain. While their results are in line with reductions seen in motor evoked responses during pain and effort was made to address possible confounding factors (study 2 and 3). This study opens the way for the use exploration of cortical excitability outside M1 in acute pain, and potentially in chronic pain instances. While there is still open discussion on the best strategy to handle auditory and mechanical tactile noise, technological and methodological improvements seen in the last years have greatly improved the signal to noise ratio of TMS-EEG.

---

## [Referee Report · Reviewer #3 (Public Review)]

The present study aims to investigate whether pain influences cortical excitability. To this end, heat pain stimuli are applied to healthy human participants. Simultaneously, TMS pulses are applied to M1 and TMS-evoked potentials (TEPs) and pain ratings are assessed after each TMS pulse. TEPs are used as measures of cortical excitability. The results show that TEP amplitudes at 45 msec (N45) after TMS pulses are higher during painful stimulation than during non-painful warm stimulation. Control experiments indicate that auditory, somatosensory, or proprioceptive effects cannot explain this effect. Considering that the N45 might reflect GABAergic activity, the results suggest that pain changes GABAergic activity. The authors conclude that TEP indices of GABAergic transmission might be useful as biomarkers of pain sensitivity.

Pain-induced cortical excitability changes is an interesting, timely, and potentially clinically relevant topic. The paradigm and the analysis are sound, the results are convincing, and the interpretation is adequate. The findings will be of interest to researchers interested in the brain mechanisms of pain.

---

## [Author Response]

The following is the authors’ response to the original reviews.

**Reviewer #1 (Public Review):**
The objective of this investigation was to determine whether experimental pain could induce alterations in cortical inhibitory/facilitatory activity observed in TMS-evoked potentials (TEPs). Previous TMS investigations of pain perception had focused on motor evoked potentials (MEPs), which reflect a combination of cortical, spinal, and peripheral activity, as well as restricting the focus to M1. The main strength of this investigation is the combined use of TMS and EEG in the context of experimental pain. More specifically, Experiment 1 investigated whether acute pain altered cortical excitability, reflected in the modulation of TEPs. The main outcome of this study is that relative to non-painful warm stimuli, painful thermal stimuli led to an increase on the amplitude of the TEP N45, with a larger increase associated with higher pain ratings. Because it has been argued that a significant portion of TEPs could reflect auditory potentials elicited by the sound (click) of the TMS, Experiment 2 constituted a control study that aimed to disentangle the cortical response related to TMS and auditory activity. Finally, Experiment 3 aimed to disentangle the cortical response to TMS and reafferent feedback from muscular activity elicited by suprathreshold TMS applied over M1. The fact that the authors accompanied their main experiment with two control experiments strengthens the conclusion that the N45 TEP peak could be implicated in the perception of painful stimuli.Perhaps, the addition of a highly salient but non-painful stimulus (i.e. from another modality) would have further ruled out that the effects on the N45 are not predominantly related to intensity/saliency of the stimulus rather than to pain per se.

We thank the reviewer for their comment on the possibility of whether stimulus intensity influences the N45 as opposed to pain per se. We agree that the ideal experiment would have included multiple levels of stimulation. We would argue, however, that that in Experiment 1, despite the same level of stimulus intensity for all participants (46 degrees), individual differences in pain ratings were associated with the change in the N45 amplitude, suggesting that the results cannot be explained by stimulus intensity, but rather by pain intensity.

**Reviewer #2 (Public Review):**
The authors have used transcranial magnetic stimulation (TMS) and motor evoked potentials (MEPs) and TMS-electroencephalography (EEG) evoked potentials (TEPs) to determine how experimental heat pain could induce alterations in these metrics.In Experiment 1 (n = 29), multiple sustained thermal stimuli were administered over the forearm, with the first, second, and third block of stimuli consisting of warm but non-painful (pre-pain block), painful heat (pain block) and warm but non-painful (post-pain block) temperatures respectively. Painful stimuli led to an increase in the amplitude of the fronto-central N45, with a larger increase associated with higher pain ratings. Experiments 2 and 3 studied the correlation between the increase in the N45 in pain and the effects of a sham stimulation protocol/higher stimulation intensity. They found that the centro-frontal N45 TEP was decreased in acute pain. The study comes from a very strong group in the pain fields with long experience in psychophysics, experimental pain, neuromodulation, and EEG in pain. They are among the first to report on changes in cortical excitability as measured by TMS-EEG over M1. While their results are in line with reductions seen in motor-evoked responses during pain and effort was made to address possible confounding factors (study 2 and 3), there are some points that need attention. In my view the most important are:1. The method used to calculate the rest motor threshold, which is likely to have overestimated its true value : calculating highly abnormal RMT may lead to suprathreshold stimulations in all instances (Experiment 3) and may lead to somatosensory "contamination" due to re-afferent loops in both "supra" and "infra" (aka. less supra) conditions.

The method used to assess motor threshold was the TMS motor threshold Assessment Tool (MTAT) which estimates motor threshold using maximum likelihood parametric estimation by sequential testing (Awiszus et al., 2003; Awiszus and Borckardt, 2011). This was developed as a quicker alternative for calculating motor threshold compared to the traditional Rossini-Rothwell method which involves determining the lowest intensity that evokes at least 5/10 MEPs of at least 50 microvolts. The method has been shown to achieve the same accuracy of determining motor threshold as the traditional Rossini-Rothwell method, but with fewer pulses (Qi et al., 2011; Silbert et al., 2013).

We have now made this clearer in the manuscript:

“The RMT was determined using the TMS motor thresholding assessment tool, which estimates the TMS intensity required to induce an MEP of 50 microvolts with a 50% probability using maximum likelihood parametric estimation by sequential testing (Awiszus, 2003; Awiszus & Borckardt, 2011). This method has been shown to achieve the accuracy of methods such as the Rossini-Rothwell method (Rossini et al., 1994; Rothwell et al., 1999) but with fewer pulses (Qi, Wu, & Schweighofer, 2011; Silbert, Patterson, Pevcic, Windnagel, & Thickbroom, 2013). The test stimulus intensity was set at 110% RMT to concurrently measure MEPs and TEPs during pre-pain, pain and post-pain blocks.”

Therefore, the high RMTs in our study cannot be explained by the threshold assessment method. Instead, they are likely explained by aspects of the experimental setup that increased the distance between the TMS coil and the scalp, including the layer of foam placed over the coil, the EEG cap and the fact that the electrodes we used had a relatively thick profile. This has been explained in the paper:

“We note that the relatively high RMTs are likely due to aspects of the experimental setup that increased the distance between the TMS coil and the scalp, including the layer of foam placed over the coil, the EEG cap and relatively thick electrodes (6mm)”

Awiszus, F. (2003). TMS and threshold hunting. In Supplements to Clinical neurophysiology (Vol. 56, pp. 13-23). Elsevier.

Qi, F., Wu, A. D., & Schweighofer, N. (2011). Fast estimation of transcranial magnetic stimulation motor threshold. Brain stimulation, 4(1), 50-57.

Silbert, B. I., Patterson, H. I., Pevcic, D. D., Windnagel, K. A., & Thickbroom, G. W. (2013). A comparison of relative-frequency and threshold-hunting methods to determine stimulus intensity in transcranial magnetic stimulation. Clinical Neurophysiology, 124(4), 708-712.

1. The low number of pulses used for TEPs (close to ⅓ of the usual and recommended)

We agree that increasing the number of pulses can increase the signal to noise ratio. During piloting, participants were unable to tolerate the painful stimulus for long periods of time and we were required to minimize the number of pulses per condition.

We note that there is no set advised number of trials in TMS-EEG research. According to the recommendations paper, the number of trials should be based on the outcome measure e.g., TEP peaks vs. frequency domain measures vs. other measures and based on previous studies investigating test-retest reliability (Hernandez-Pavon et al., 2023). The choice of 66 pulses per condition was based on the study by Kerwin et al., (2018) showing that optimal concordance between TEP peaks can be found with 60-100 TMS pulses delivered in the same run (as in the present study). The concordance was particularly higher for the N40 peak at prefrontal electrodes, which was the key peak and electrode cluster in our study. We have made this clearer:

“Current recommendations (Hernandez-Pavon et al., 2023) suggest basing the number of TMS trials per condition on the key outcome measure (e.g., TEP peaks vs. frequency measures) and based on previous test-retest reliability studies. In our study the number of trials was based on a test-retest reliability study by (Kerwin, Keller, Wu, Narayan, & Etkin, 2018) which showed that 60 TMS pulses (delivered in the same run) was sufficient to obtain reliable TEP peaks (i.e., sufficient within-individual concordance between the resultant TEP peaks of each trial).”

Further supporting the reliability of the TEP data in our experiment, we note that the scalp topographies of the TEPs for active TMS at various timepoints (Figures 5, 7 and 9) were similar across all three experiments, especially at 45 ms post-TMS (frontal negative activity, parietal-occipital positive activity).

In addition to this, the interclass correlation coefficient (Two-way fixed, single measure) for the N45 to active suprathreshold TMS across timepoints for each experiment was 0.90 for Experiment 1 (across pre-pain, pain, post-pain time points), 0.74 for Experiment 2 (across pre-pain and pain conditions), and 0.95 for Experiment 3 (across pre-pain conditions). This suggests that even with the fluctuations in the N45 induced by pain, the N45 for each participant was stable across time, further supporting the reliability of our data. These ICCs are now reported in the supplementary material (subheading: Test-retest reliability of N45 Peaks).

Hernandez-Pavon, J. C., Veniero, D., Bergmann, T. O., Belardinelli, P., Bortoletto, M., Casarotto, S., ... & Ilmoniemi, R. J. (2023). TMS combined with EEG: Recommendations and open issues for data collection and analysis. Brain Stimulatio, 16(3), 567-593

Kerwin, L. J., Keller, C. J., Wu, W., Narayan, M., & Etkin, A. (2018). Test-retest reliability of transcranial magnetic stimulation EEG evoked potentials. Brain stimulation, 11(3), 536-544.

Lack of measures to mask auditory noise.

In TMS-EEG research, various masking methods have been proposed to suppress the somatosensory and auditory artefacts resulting from TMS pulses, such as white noise played through headphones to mask the click sound (Ilmoniemi and Kičić, 2010), and a thin layer of foam placed between the TMS coil and EEG cap to minimize the scalp sensation (Massimini et al., 2005). However, recent studies have shown that even when these methods are used, sensory contamination of TEPs is still present, as shown by studies that show commonalities in the signal between active and sensory sham conditions that mimic the auditory/somatosensory aspects of real TMS (Biabani et al., 2019; Conde et al., 2019; Rocchi et al., 2021). This has led many authors (Biabani et al., 2019; Conde et al., 2019) to recommend the use of sham conditions to control for sensory contamination. To separate the direct cortical response to TMS from sensory evoked activity, Experiment 2 included a sham TMS condition that mimicked the auditory/somatosensory aspects of active TMS to determine whether any alterations in the TEP peaks in response to pain were due to changes in sensory evoked activity associated with TMS, as opposed to changes in cortical excitability. Therefore, the lack of auditory masking does not impact the main conclusions of the paper.

We have made this clearer:

“… masking methods have been used to suppress these sensory inputs, (Ilmoniemi and Kičić, 2010; Massimini et al., 2005). However recent studies have shown that even when these methods are used, sensory contamination of TEPs is still present, as shown by commonalities in the signal between active and sensory sham conditions that mimic the auditory/somatosensory aspects of real TMS (Biabani et al., 2019; Conde et al., 2019; Rocchi et al., 2021). This has led many leading authors (Biabani et al., 2019; Conde et al., 2019) to recommend the use of sham conditions to control for sensory contamination.”

Ilmoniemi, R. J., & Kičić, D. (2010). Methodology for combined TMS and EEG. Brain topography, 22, 233-248.

Massimini, M., Ferrarelli, F., Huber, R., Esser, S. K., Singh, H., & Tononi, G. (2005). Breakdown of cortical effective connectivity during sleep. Science, 309(5744), 2228-2232.

Biabani, M., Fornito, A., Mutanen, T. P., Morrow, J., & Rogasch, N. C. (2019). Characterizing and minimizing the contribution of sensory inputs to TMS-evoked potentials. Brain stimulation, 12(6), 1537-1552.

Conde, V., Tomasevic, L., Akopian, I., Stanek, K., Saturnino, G. B., Thielscher, A., ... & Siebner, H. R. (2019). The non-transcranial TMS-evoked potential is an inherent source of ambiguity in TMS-EEG studies. Neuroimage, 185, 300-312.

Rocchi, L., Di Santo, A., Brown, K., Ibáñez, J., Casula, E., Rawji, V., ... & Rothwell, J. (2021). Disentangling EEG responses to TMS due to cortical and peripheral activations. Brain stimulation, 14(1), 4-18.

1. A supra-stimulus heat stimulus not based on individual HPT, that oscillates during the experiment and that lead to large variations in pain intensity across participants is unfortunate.

The choice of whether to calibrate or fix stimulus intensity is a contentious question in experimental pain research. A recent discussion by Adamczyk et al., (2022) explores the pros and cons of each approach and recommends situations where one method may be preferred over the other. That paper suggests that the choice of the methodology is related to the research question – when the main outcome of the research is objective (neurophysiological measures) and researchers are interested in the variability in pain ratings, the fixed approach is preferrable. Given we explored the relationship between MEP/N45 modulation by pain and pain intensity, this question is better explored by using the same stimulus intensity for all participants, as opposed to calibrating the intensity to achieve a similar level of pain across participants.

We have made this clearer:

“Given we were interested in the individual relationship between pain and excitability changes, the fixed temperature of 46ºC ensured larger variability in pain ratings as opposed to calibrating the temperature of the thermode for each participant (Adamczyk et al., 2022).”.

Adamczyk, W. M., Szikszay, T. M., Nahman-Averbuch, H., Skalski, J., Nastaj, J., Gouverneur, P., & Luedtke, K. (2022). To calibrate or not to calibrate? A methodological dilemma in experimental pain research. The Journal of Pain, 23(11), 1823-1832.

So is the lack of report on measures taken to correct for a fortuitous significance (multiple comparison correction) in such a huge number of serial paired tests.

Note that we used a Bayesian approach for all analyses as opposed to the traditional frequentist approach. In contrast to the frequentist approach, the Bayesian approach does not require corrections for multiple comparisons (Gelman et al., 2000) given that they provide a ratio representing the strength of evidence for the null vs. alternative hypotheses as opposed to accepting or rejecting the null hypothesis based on p-values. As such, throughout the paper, we frame our interpretations and conclusions based on the strength of evidence (e.g. anecdotal/weak, moderate, strong, very strong) as opposed to referring to the significance of the effects.

Gelman A, Tuerlinckx F. (2000). Type S error rates for classical and Bayesian single and multiple comparison procedures. Computational statistics, 15(3):373-90.

**Reviewer #3 (Public Review):**
The present study aims to investigate whether pain influences cortical excitability. To this end, heat pain stimuli are applied to healthy human participants. Simultaneously, TMS pulses are applied to M1 and TMS-evoked potentials (TEPs) and pain ratings are assessed after each TMS pulse. TEPs are used as measures of cortical excitability. The results show that TEP amplitudes at 45 msec (N45) after TMS pulses are higher during painful stimulation than during non-painful warm stimulation. Control experiments indicate that auditory, somatosensory, or proprioceptive effects cannot explain this effect. Considering that the N45 might reflect GABAergic activity, the results suggest that pain changes GABAergic activity. The authors conclude that TEP indices of GABAergic transmission might be useful as biomarkers of pain sensitivity.Pain-induced cortical excitability changes is an interesting, timely, and potentially clinically relevant topic. The paradigm and the analysis are sound, the results are mostly convincing, and the interpretation is adequate. The following clarifications and revisions might help to improve the manuscript further.1. Non-painful control condition. In this condition, stimuli are applied at warmth detection threshold. At this intensity, by definition, some stimuli are not perceived as different from the baseline. Thus, this condition might not be perfectly suited to control for the effects of painful vs. non-painful stimulation. This potential confound should be critically discussed.

In Experiment 3, we also collected warmth ratings to confirm whether the pre-pain stimuli were perceived as different from baseline. This detail has been added to them methods:

“In addition to the pain rating in between TMS pulses, we collected a second rating for warmth of the thermal stimulus (0 = neutral, 10 = very warm) to confirm that the participants felt some difference in sensation relative to baseline during the pre-pain block. This data is presented in the supplementary material”.

We did not include these data in the initial submission but have now included it in the supplemental material. These data showed warmth ratings were close to 2/10 on average. This confirms that the non-painful control condition produced some level of non-painful sensation.

1. MEP differences between conditions. The results do not show differences in MEP amplitudes between conditions (BF 1.015). The analysis nevertheless relates MEP differences between conditions to pain ratings. It would be more appropriate to state that in this study, pain did not affect MEP and to remove the correlation analysis and its interpretation from the manuscript.

The interindividual relationship between changes in MEP amplitude and individual pain rating is statistically independent from the overall group level effect of pain on MEP amplitude. Therefore, conclusions for the individual and group level effects can be made independently.

It is also important to note that in the pain literature, there is now increasing emphasis placed on investigating the individual level relationship between changes in cortical excitability and pain as opposed to the group level effect (Seminowicz et al., 2019; Summers et al., 2019). As such, it is important to make these results readily available for the scientific community.

We have made this clearer:

‘As there is now increasing emphasis placed on investigating the individual level relationship between changes in cortical excitability and pain and not only the group level effect, (Chowdhury et al., 2022; Seminowicz et al., 2018; Seminowicz, Thapa, & Schabrun, 2019; Summers et al., 2019) we also investigated the correlations between pain ratings and changes in MEP (and TEP) amplitude”

Chowdhury, N. S., Chang, W. J., Millard, S. K., Skippen, P., Bilska, K., Seminowicz, D. A., & Schabrun, S. M. (2022). The Effect of Acute and Sustained Pain on Corticomotor Excitability: A Systematic Review and Meta-Analysis of Group and Individual Level Data. The Journal of Pain, 23(10), 1680-1696.

Summers, S. J., Chipchase, L. S., Hirata, R., Graven-Nielsen, T., Cavaleri, R., & Schabrun, S. M. (2019). Motor adaptation varies between individuals in the transition to sustained pain. Pain, 160(9), 2115-2125.

Seminowicz, D. A., Thapa, T., & Schabrun, S. M. (2019). Corticomotor depression is associated with higher pain severity in the transition to sustained pain: a longitudinal exploratory study of individual differences. The Journal of Pain, 20(12), 1498-1506.

1. Confounds by pain ratings. The ISI between TMS pulses is 4 sec and includes verbal pain ratings. Considering this relatively short ISI, would it be possible that verbal pain ratings confound the TEP? Moreover, could the pain ratings confound TEP differences between conditions, e.g., by providing earlier ratings when the stimulus is painful? This should be carefully considered, and the authors might perform control analyses.

It is unlikely that the verbal ratings contaminated the TEP response as the subsequent TMS pulse was not delivered until the verbal rating was complete and given that each participant was cued by the experimenter to provide the pain rating after each pulse (rather than the participant giving the rating at any time). As such, it would not be possible for participants to provide earlier ratings to more painful stimuli.

We have made this clearer:

"To avoid contamination of TEPs by verbal ratings, the subsequent TMS pulse was not delivered until the verbal rating was complete, and the participant was cued by the experimenter to provide the pain rating after each pulse.”

1. Confounds by time effects. Non-painful and painful conditions were performed in a fixed order. Potential confounds by time effects should be carefully considered.

Previous research suggests that pain alters neural excitability even after pain has subsided. In a recent meta-analysis (Chowdhury et al., 2022) we found effect sizes of 0.55-0.9 for MEP reductions 0-30 minutes after pain had resolved. As such, we avoided intermixing pain and warm blocks given subsequent warm blocks would not serve as a valid baseline, as each subsequent warm block would have residual effects from the previous pain blocks.

Chowdhury, N. S., Chang, W. J., Millard, S. K., Skippen, P., Bilska, K., Seminowicz, D. A., & Schabrun, S. M. (2022). The Effect of Acute and Sustained Pain on Corticomotor Excitability: A Systematic Review and Meta-Analysis of Group and Individual Level Data. The Journal of Pain, 23(10), 1680-1696.

At the same time, given there was no conclusive evidence for a difference in N45 amplitude between pre-pain and post-pain conditions of Experiment 1 (Supplementary Figure 1), it is unlikely that the effect of pain was an artefact of time i.e., the explanation that successive thermal stimuli applied to the skin results an increase in the N45, regardless of whether the stimuli are painful or not. We will make this point in our next revision.

We have discussed this issue:

“Lastly, future research should consider replicating our experiment using intermixed pain and no pain blocks, as opposed to fixed pre-pain and pain blocks, to control for order effects i.e., the explanation that successive thermal stimuli applied to the skin results an increase in the N45 peak, regardless of whether the stimuli are painful or not. However, we note that there was no conclusive evidence for a difference in N45 peak amplitude between pre-pain and post-pain conditions of Experiment 1 (Supplementary Figure 1), suggesting it is unlikely that the observed effects were an artefact of time.”

1. Data availability. The authors should state how they make the data openly available.

We have uploaded the MEP, TEP and pain data on the Open science framework https://osf.io/k3psu/

**Reviewer #1 (Recommendations For The Authors):**
I think the study is quite solid and I only have very minor recommendations for the authors:Introduction, p. 3: "Functional magnetic resonance imaging has helped us understand where in the brain pain is processed". This is an overstatement. fMRI provides us with potential biomarkers (e.g. "the pain signature"), but the specificity of these responses for pain is debated and we still do not know where in the brain pain is processed.

We have amended to:

“functional magnetic resonance imaging has assisted in the localization of brain structures implicated in pain processing”

Introduction, p. 5: "neural baseline" should be "neutral baseline"?

We thank the reviewer for identifying this – this has now been amended.

**Reviewer #2 (Recommendations For The Authors):**
INTRODUCTIONThe introduction mentions how important extra-motor areas can be explored by TMS-EEG, then the effects of DLPFC rTMS on TEPs ... but you do not explore the DLPFC... Perhaps the introduction should be reframed.

The current work explores cortical excitability throughout the brain (as shown in our cluster-based permutation and source localization analyses), so our investigations are in line with the introductions statement about the importance of studying non-motor areas.

The reference to DLPFC rTMS was to highlight current existing research that has applied TMS-EEG to understand pain. It was not used as a methodological rationale to investigate the DLPFC in the present study. To make the research gap clearer, we state:

“While these studies assist us in understanding whether TEPs might mediate rTMS-induced pain reductions, no study has investigated whether TEPs are altered in direct response to pain”

Lignes 63-65 the term "TMS" is used to refer to motor corticospinal excitability measures, in contrast to TMS-EEG measures of TEPs. Then the authors come back to TMS-EEG and then again back to MEPs. This is rather confusing: TMS means TMS... the concept of MEP/ motor corticospinal excitability measures is not intuitive when using the term "TMS". I suggest using motor corticospinal excitability measures when referring to MEP/MEP-based measures of cortical excitability... and M1TMS-EEG-evoked potentials (usually abbreviated to TEPs) to refer to TMS-EEG responses as measured here.

Throughout the manuscript, we now use the term TEPs when referring to TMS-EEG measures, and MEPs when referring to TMS-EMG measure. The use of TEPs vs. MEPs will make it easier for readers to follow which measures we are referring to.

Line 83: "As such, the precise origin of the pain mechanism cannot be localized." Please rephrase, the sentence conveys the idea that it is indeed possible to localize the origin of a pain mechanism with a different approach, and we know this is not currently possible, irrespective of the methodological setup.

We have replaced this with:

“This makes it unclear as to whether pain processes occur at the cortical, spinal or peripheral level.”

How can one predetermine the temperature that will be perceived as painful by someone else, and not base it on individual HPT? This is against principles of psychophysics. Please comment. Attesting all participants had HPT below 46 is important, but then being stimulated at 46C when our HPT is 45C is different from when our HPT is 39C. Please explain why the pain intensity was not standardised based on individual HPT.

Please refer to our response to the public review related to the issue

Line 38: "if we had used an alternative design with blocks of warm stimuli intermixed with blocks of painful stimuli, the warm stimuli blocks would not serve as a valid non-painful baseline". I do not understand why it is not possible to have a pain-free baseline, followed by a pain/warm sequence.

In our study, we had the choice of either intermixing blocks or to use a fixed sequence. Previous research suggests that pain alters neural excitability even after pain has subsided. In a recent meta-analysis (Chowdhury et al., 2022) we found effect sizes of 0.55-0.9 for MEP reductions 0-30 minutes after pain had resolved. As such, we avoided intermixing pain and warm blocks given subsequent warm blocks would not serve as a valid baseline, as each subsequent warm block would have residual effects from the previous pain blocks.

We have updated the manuscript to be clearer about why we used a fixed sequence:

“The pre-pain/pain/post-pain design has been commonly used in the TMS-MEP pain literature, as many studies have demonstrated strong changes in corticomotor excitability that persist beyond the painful period. Indeed, in a systematic review, we showed effect sizes of 0.55-0.9 for MEP reductions 0-30 minutes after pain had resolved (Chowdhury et al., 2022). As such, if we had used an alternative design with blocks of warm stimuli intermixed with blocks of painful stimuli, the warm stimuli blocks would not serve as a valid non-painful baseline”

Chowdhury, N. S., Chang, W. J., Millard, S. K., Skippen, P., Bilska, K., Seminowicz, D. A., & Schabrun, S. M. (2022). The Effect of Acute and Sustained Pain on Corticomotor Excitability: A Systematic Review and Meta-Analysis of Group and Individual Level Data. The Journal of Pain, 23(10), 1680-1696.

Please explain, and provide evidence that stimulation of people with predetermined temperatures is able to create warm/pain/warm sensations, without entraining pain in the last warm stimulation.

A previous study by Dube et al. (2011) used sequences of warm (36°C), painful and neutral (32° C) and found that participants did not experience pain at any time when the temperature was at a warm temperature of 36°C. We have now cited this study:

“Based on a previous study (Dubé & Mercier, 2011) which also used sequences of painful (50ºC) and warm (36°C) thermal stimuli, we did not anticipate that the stimulus in the pain block would entrain pain in the post-pain block”

Dubé, J. A., & Mercier, C. (2011). Effect of pain and pain expectation on primary motor cortex excitability. Clinical neurophysiology, 122(11), 2318-2323.

METHODSIt is not clear if participants with chronic pain, present in 20% of the general population, were excluded. If they were, please provide "how" in methods.

We excluded participants with a history or presence of acute/chronic pain. This has now been clarified:

“Participants were excluded if they had a history of chronic pain condition or any current acute pain”

Line 489: the definition of warm detection threshold is unusual, please provide a reference.

We used an identical method to Furman et al., (2020). We have made the reference to this clearer:“Warmth, cold and pain thresholds were assessed in line with a previous study (Furman et al., 2020)”

Furman, A. J., Prokhorenko, M., Keaser, M. L., Zhang, J., Chen, S., Mazaheri, A., & Seminowicz, D. A. (2020). Sensorimotor peak alpha frequency is a reliable biomarker of prolonged pain sensitivity. Cerebral Cortex, 30(12), 6069-6082.

In Experiment 2, please explain how the lack of randomisation between "pre-pain" and "pain" may have influenced results.

Given we tried to replicate Experiment 1’s methodology as close as possible (to isolate the source of the effect from Experiment 1) we chose to repeat the same sequence of blocks as Experiment 1: pre-pain followed by pain.

Given there was no conclusive evidence for a difference in N45 amplitude between pre-pain and post-pain conditions of Experiment 1 (Supplementary Figure 1), it is unlikely that the effect of pain was an order effect i.e., the explanation that successive thermal stimuli applied to the skin results an increase in the N45, regardless of whether the stimuli are painful or not.

We now discuss the issue of randomization:

“Lastly, future research should consider replicating our experiment using intermixed pain and no pain blocks, as opposed to fixed pre-pain and pain blocks, to control for order effects i.e. the explanation that successive thermal stimuli applied to the skin results an increase in the N45 peak, regardless of whether the stimuli are painful or not. However, we note that there was no conclusive evidence for a difference in N45 peak amplitude between pre-pain and post-pain conditions of Experiment 1 (Supplementary Figure 1), suggesting it is unlikely that the observed effects were an artefact of time”

Also, in Methods in general, disclose how pain intensity was assessed, and how.

Pain intensity was assessed using a verbal rating scale (0 = no pain, and 10 = most pain imaginable). We have provided more detail:

“During each 40 second thermal stimulus, TMS pulses were manually delivered, with a verbal pain rating score (0 = no pain, and 10 = worst pain imaginable) obtained between pulses. To avoid contamination of TEPs by verbal ratings, the subsequent TMS pulse was not delivered until the verbal rating was complete, and the participant was cued by the experimenter to provide the pain rating after each pulse”

Please explain how auditory masking was made during data collection.

Auditory masking noise was not played through the headphones, given that Experiment 2 controlled for auditory evoked potentials. We have made this clearer:

“Auditory masking was not used. Instead, auditory evoked potentials resulting from the TMS click sound were controlled for in Experiment 2”

Please explain if online TEP monitoring was used during data collection

Online TEP monitoring was not available with our EEG software. We have made this clearer in the manuscript:

“Online TEP monitoring was not available with the EEG software”

Line 499: what is subthreshold TMS here? You are measuring TEPs, and not MEPs initially, so you may have a threshold for MEPs and TEPs, which are not the same.

The intensity was calibrated relative to the MEP response (rather than TEP response) - this has now been clarified:

“… and the inclusion of a subthreshold TMS (90% of resting motor threshold) condition intermixed within both the pre-pain and pain blocks.”

Please provide a reference and a figure to illustrate the electric stimulation used in the sham procedure in Study 2

The apparatus for the electrical stimulation is shown in Figure 7A, and was based on previous papers using electrical stimulation over motor cortex to simulate the somatosensory aspect of real TMS (Chowdhury et al., 2022; Gordon et al., 2022; Rocchi et al., 2021). We have made this clearer:

“Electrical stimulation was based on previous studies attempting to simulate the somatosensory component of active TMS (Chowdhury et al., 2022; Gordon et al., 2022; Rocchi et al., 2021)”

Gordon, P. C., Jovellar, D. B., Song, Y., Zrenner, C., Belardinelli, P., Siebner, H. R., & Ziemann, U. (2021). Recording brain responses to TMS of primary motor cortex by EEG–utility of an optimized sham procedure. Neuroimage, 245, 118708.

Chowdhury, N. S., Rogasch, N. C., Chiang, A. K., Millard, S. K., Skippen, P., Chang, W. J., ... & Schabrun, S. M. (2022). The influence of sensory potentials on transcranial magnetic stimulation–Electroencephalography recordings. Clinical Neurophysiology, 140, 98-109.

Rocchi, L., Di Santo, A., Brown, K., Ibánez, J., Casula, E., Rawji, V., ... & Rothwell, J. (2021). Disentangling EEG responses to TMS due to cortical and peripheral activations. Brain stimulation, 14(1), 4-18.

It is not so common to use active electrodes for TMS-EEG. Please confirm the electrodes used and if they are c-ring TMS compatible and provide reference if otherwise (or actual papers recommending active ones)

To be more specific about the electrode type we have indicated:

“Signals were recorded from 63 TMS-compatible active electrodes (6mm height, 13mm width), embedded in an elastic cap (ActiCap, Brain Products, Germany), in line with the international 10-10 system”

A paper directly comparing TEPs between active and passive electrodes found no difference between the two and concluded TEPs can be reliably obtained using active electrodes (Mancuso et al., 2021). There is also evidence that active electrodes have better signal quality than passive electrodes at higher impedance levels (Laszlo et al., 2014).

This information has now been added to the paper:

“Active electrodes result in similar TEPs (both magnitude and peaks) to more commonly used passive electrodes (Mancuso et al., 2021). There is also evidence that active electrodes have higher signal quality than passive electrodes at higher impedance levels (Laszlo, Ruiz-Blondet, Khalifian, Chu, & Jin, 2014).”

There is a growing literature showing that monophonic pulses are not reliable for TEPs when compared to biphasic ones, please provide references. https://doi.org/10.1016/j.brs.2023.02.009

The reference provided by the reviewer states that biphasic and monophasic pulses both have advantages and disadvantages, rather than stating “monophonic pulses are not reliable for TEPs”. While there is some evidence that the artefacts resulting from monophasic pulses are larger than biphasic pulses, the EEG signal still returns to baseline levels within 5ms of the TMS pulse (Rogasch et al., 2013). Moreover, one paper (Casula et al. 2018) found that the resultant TEPs evoked by monophasic pulses are larger than those resulting from biphasic pulses. The authors postulated that monophasic pulses are more effective at activating widespread cortical areas than biphasic pulses. Ultimately the reference provided by the reviewer concludes that “effect of pulse shape on TEPs has not been systematically investigated and more studies are needed”.

Rogasch, N. C., Thomson, R. H., Daskalakis, Z. J., & Fitzgerald, P. B. (2013). Short-latency artifacts associated with concurrent TMS–EEG. Brain stimulation, 6(6), 868-876.

Casula, E. P., Rocchi, L., Hannah, R., & Rothwell, J. C. (2018). Effects of pulse width, waveform and current direction in the cortex: A combined cTMS-EEG study. Brain stimulation, 11(5), 1063-1070.

In most heads, a pulse in the PA direction is not obtained by a coil oriented 45o to the midline. The later induced later-medial pulses, good to obtain MEPs

We followed previous studies measuring MEPs from the ECRB elbow muscle (Schabrun et al., 2016; de Martino et al., 2019) whereby the TMS coil handle was angled at 45 degrees relative to the midline in order to induce a posterior-anterior current. We are not aware of literature that shows that the 45 degrees orientation does not induce a posterior anterior current in most heads.

Schabrun, S. M., Christensen, S. W., Mrachacz-Kersting, N., & Graven-Nielsen, T. (2016). Motor cortex reorganization and impaired function in the transition to sustained muscle pain. Cerebral Cortex, 26(5), 1878-1890.

De Martino, E., Seminowicz, D. A., Schabrun, S. M., Petrini, L., & Graven-Nielsen, T. (2019). High frequency repetitive transcranial magnetic stimulation to the left dorsolateral prefrontal cortex modulates sensorimotor cortex function in the transition to sustained muscle pain. Neuroimage, 186, 93-102.

The definition of RMT is (very) unusual. RMT provides small 50microV MEPs in 50% of times. If you obtain MEPs at 50microV you are supra threshold!

The TMS motor threshold assessment tool calculates threshold in the same manner as other threshold tools – it calculates the intensity that elicits an MEP of 50 microvolts, 50% of the time. We have made this clearer:

“The RMT was determined using the TMS motor thresholding assessment tool, which estimates the TMS intensity required to induce an MEP of 50 microvolts with a 50% probability using maximum likelihood parametric estimation by sequential testing (Awiszus and Borckardt, 2011). This method has been shown to achieve the accuracy of methods such as the Rossini-Rothwell method (Rossini et al., 1994; Rothwell et al., 1999) but with fewer pulses (Qi et al., 2011; Silbert et al., 2013).”

Please inform the inter TMS pulse interval used of TEPs and whether they were randomly generated.

The pulses were delivered manually – the interval was not randomly generated – as stated:

“As TMS was delivered manually, there was no set interpulse interval. However, the 40 second stimulus duration allowed for 11 pulses for each heat stimulus …. (~ 4 seconds in between …)”

Why have you stimulated suprathreshold on M1 when assessing TEP´s? The whole idea is that large TEPs can be obtained at lower intensities below real RMT and that prevents re-entering loops of somatosensory and joint movement inputs that insert "noise" to the TEPs.

The suprathreshold intensity was used to concurrently measure MEPs during pre-pain, pain and post-pain blocks.

We have made this clearer:

“The test stimulus intensity was set at 110% RMT to concurrently measure MEPs and TEPs during pre-pain, pain and post-pain blocks.”

The influence of re-afferent muscle activity was controlled for in Experiment 3.

Did you assess pain intensity after each of the TEP pulses? Please discuss how such a cognitive task may have influenced results

Pain intensity was assessed after each TMS pulse, as stated:

“TMS pulses were manually delivered, with a verbal pain rating score (0 = no pain, and 10 = most pain imaginable) obtained between pulses”

Reviewer 3 also brought up a concern of whether the verbal rating task might have influenced the TEPs. However, it is unlikely that the task contaminated the TEP response as the subsequent TMS pulse was not delivered until the verbal rating was complete and given that each participant was cued by the experimenter to provide the pain rating after each pulse (rather than the participant giving the rating at any time). We have made this clearer where we state:

“To avoid contamination of TEPs by verbal ratings, the subsequent TMS pulse was not delivered until the verbal rating was complete, and the participant was cued by the experimenter to provide the pain rating after each pulse”

The QST approach is unusual. Please confirm the sequence of CDT, WDT and HPT were not randomised and that no interval beyond 6sec were used. Proper references are welcome.

In line with a previous study (Furman et al., 2020), the sequence of the CPT, WDT and HPT were not randomized, and the interval was not more than 6 seconds.

We have made this clearer:

“A total of three trials was conducted for each test to obtain an average, with an interstimulus interval of six seconds. The sequence of cold, warmth and pain threshold was the same for all participants (Furman et al. 2020)”

Performing 60 pulses for TEPs is unusual, and against the minimum number in recommendationsPlease explain and comment.https://doi.org/10.1016/j.brs.2023.02.009

Please refer to our previous response to this concern in the public reviews.

Line 578: when you refer to "heat" the reader may confound warm/heat with heat meaning suprathreshold. Please revise the wording.

We have now replaced the word heat stimulus with thermal stimulus.

Why were Bayesian statistics used instead as frequentist ones?

We have made this clearer:

“Given we were interested in determining the evidence for pain altering TEP peaks in certain conditions (e.g., active TMS) and pain not altering TEP peaks in other conditions (sham TMS), we used a Bayesian approach as opposed to a frequentist approach, which considers the strength of the evidence for the alternative vs. null hypothesis”

RESULTSThere is a huge response with high power after 100ms- Please discuss if you believe auditory potentials may have influenced it.

It is indeed possible that auditory potentials were present at 100ms. We now state:

“Indeed, the signal at ~100ms post-TMS from Experiment 1 may reflect an auditory N100 response”

The presence of auditory contamination does not impact the main conclusions of the paper given this was controlled for in Experiment 2.

Please discuss how pain ranging from 3-10 may have influenced results in the "PAIN" situation,

It is anticipated that the fixed thermal stimulus intensity approach would lead to large variations in pain ratings (Adamczyk et al., 2022). This is a recommended approach when the aim of the research is to determine relationships between neurophysiological measures and individual differences in pain sensitivity (Adamczyk et al., 2022). Indeed, we were interested in whether alterations in neurophysiological measures were associated with pain intensity, and we found that higher pain ratings were associated with smaller reductions in MEP amplitude and larger increases in N45 amplitude.

Adamczyk, W. M., Szikszay, T. M., Nahman-Averbuch, H., Skalski, J., Nastaj, J., Gouverneur, P., & Luedtke, K. (2022). To calibrate or not to calibrate? A methodological dilemma in experimental pain research. The Journal of Pain, 23(11), 1823-1832.

Please indicate if any participants offered pain after warm stimulation ( possible given secondary hyperalgesia after so many plateaux of heat stimulation).

As stated in the results “All participants reported 0/10 pain during the pre-pain and post-pain blocks”.

Please discuss the potential effects of having around 10% of "bad channels In average per experiment per participants, its impacts in source localisation and in TEP measurement. Same for >5 epochs excluded by participant.

The number of bad channels has been incorrectly stated by the reviewer as being 10% on average per experiment per participant, whereas the correct number of reported bad channels was 3%, 4.7% and 9.8% for Experiment 1, 2 and 3 respectively (see supplementary material). These numbers are below the accepted number of bad channels to interpolate (10%) in EEG pipelines (e.g., Debnath et al., 2020; Kayhan et al., 2022), so it is unlikely that our channel exclusions significantly influenced the quality of our source localization an TEP data.

Debnath, R., Buzzell, G. A., Morales, S., Bowers, M. E., Leach, S. C., & Fox, N. A. (2020). The Maryland analysis of developmental EEG (MADE) pipeline. Psychophysiology, 57(6), e13580.

Kayhan, E., Matthes, D., Haresign, I. M., Bánki, A., Michel, C., Langeloh, M., ... & Hoehl, S. (2022). DEEP: A dual EEG pipeline for developmental hyperscanning studies. Developmental cognitive neuroscience, 54, 101104.

The number of excluded epochs is unlikely to have influenced the results given there was evidence for no difference in the number of rejected epochs between conditions (E1 BF10 = 0.145, E2 BF10 = 0.27, E3 BF10 = 0.169 – these BFs have now been reported in the supplementary material), and given the reliability of the N45 was high (see response to previous comment on the number of trials per condition).

HPT of 42.9 {plus minus} 2.5{degree sign}C means many participants had HPT close to 46oC. Please discuss

While some participants did indeed have pain thresholds close to 46 degrees, they nonetheless reported pain during the test blocks. While such participants may have reported less pain compared to others, we aimed for larger variations in pain ratings, given one of the research questions was to determine why pain intensity differs between individuals (given the same noxious stimulus). Indeed, we showed that this variation was meaningful (pain intensity was related to alterations in N45 and MEP amplitude).

Please explain the sentence : line 139 "As such, if we had used an alternative design with blocks of warm stimuli intermixed with blocks of painful stimuli, the warm stimuli blocks would not serve as a valid non-painful baseline." I cannot see why.

Please refer to our previous point on why the fixed sequence was included.

And on the top of that heat was not individualised according to HPT.

Please refer to our previous point on why we used a fixed stimulus approach.

Sequences of warm/heat were not randomised.Please refer to our previous point on the why the sequence of blocks was not randomized.Line 197: "However, as this is the first study investigating the effects of experimental pain on TEPsamplitude, there were no a priori regions or timepoints of interest to compare betweenconditions". This is not clear. It means you have not measured the activity (size of the N45) under the electrode closest to the TMS coil? The TEP is supposed to by higher under the stimulated target/respective corresponding electrode…

We are not aware of any current recommendations that state that the region of interest should be based on the site of stimulation. The advantage of TMS-EEG is that it allows characterisation of cortical excitability changes throughout the brain, not just the site of stimulation. We based our region of interest on a cluster-based permutation analysis, as recommended by Frömer, Maier, & Abdel Rahman, (2018)

Frömer, R., Maier, M., & Abdel Rahman, R. (2018). Group-level EEG-processing pipeline for flexible single trial-based analyses including linear mixed models. Frontiers in neuroscience, 12, 48.

Please explain where N45 values came from.

The N45 was calculated using the TESA peak function (Rogasch et al., 2017) which identifies a data point which is larger/smaller than +/- 5 data points within a specified time window (e,g, 40-70ms post-TMS as in the present study). Where multiple peaks are found, the amplitude of the largest peak is returned. Where no peak is found, the amplitude at the specified latency is returned.

Rogasch, N. C., Sullivan, C., Thomson, R. H., Rose, N. S., Bailey, N. W., Fitzgerald, P. B., ... & Hernandez-Pavon, J. C. (2017). Analysing concurrent transcranial magnetic stimulation and electroencephalographic data: A review and introduction to the open-source TESA software. Neuroimage, 147, 934-951.

If only the cluster assessment was made please provide the comparison between P45 from the target TMS channel location in pre pain vs pain.

We assume the reviewer is referring to the N45 rather than P45, and that by “target” TMS channel they are referring to the stimulated region.

We first clarify that there is no “target” channel given the motor hotspot differs between individuals and so the channel that is closest to the site of stimulation will always differ.

Secondly, as stated above, we are not aware of any current recommendations in TMS-EEG research that states that the region of interest for TEP analysis should be based on the site of stimulation. The advantage of TMS-EEG is that it allows characterisation of cortical excitability throughout the brain, not just the site of stimulation. If we based our ROI on the target channel only, we would lose valuable information about excitability changes occurring in other brain regions.

Lastly, the N45 was localized at frontocentral electrodes, which is also where the cluster differences emerged. As such, we do not believe it would be informative to compare N45 peak amplitude at the region of stimulation.

Also explain how correction for multiple comparisons was made

Please refer to our response to the public review related to this issue.

And report data from pain vs post-pain.

The pain vs. post-pain comparisons are now reported in the Supplementary material.

There is a strong possibility the response at N85 is an auditory /muscle signal. Please provide the location of this response.

We have opted not to include the topography at 85ms in the main paper as it would introduce too much clutter into the figures (which are already very dense), and because the topography was very similar to the topography at 100ms. As an example, for the reviewer, in Author response image 1 we have shown the topography for the pre-pain condition of Experiment 1.

**Author response image 1. sa4fig1:** 

Experiment 2: I have a strong impression both active TEPs and sham TEPs were contaminated by auditory (and muscle) noise. Please explain.

While it possible that auditory noise may have influenced TEPs in the active and sham groups, it does not impact the main conclusions of the paper, given that the purpose of the sham condition was to control for auditory and somatosensory stimulation resulting from TMS.

While muscle activity may also affect have influenced the TEPs in active and sham conditions, we used fastICA in all conditions to suppress muscle activity. The fastICA algorithm (Rogasch et al., 2017) runs an independent component analysis on the data, and classifies components as neural, TMS-evoked muscle, eye movements and electrode noise, based on a set of heuristic thresholding rules (e.g., amplitude, frequency and topography of the components). Components classified as TMS-evoked muscle/other muscle artefacts are then removed. In the supplementary material, we further report that the number of components removed did not differ between conditions, suggesting the impact of muscle artefacts are not larger in some conditions vs. others.

Rogasch, N. C., Sullivan, C., Thomson, R. H., Rose, N. S., Bailey, N. W., Fitzgerald, P. B., ... & Hernandez-Pavon, J. C. (2017). Analysing concurrent transcranial magnetic stimulation and electroencephalographic data: A review and introduction to the open-source TESA software. Neuroimage, 147, 934-951.

Experiment 3: One interpretation can be that both supra and sub-threshold TMS were leading to somatosensory re-afferent responses, based on the way RMT was calculated, which hyper estimate the RMT and delivers in reality 2 types of supra-threshold stimulations. Please discuss

Please refer to our response to the public review related to this issue.

Please provide correlation between N45 size and MEPs amplitudes.

This has now been included:

“There was no conclusive evidence of any relationship between alterations in MEP amplitude during pain, and alterations in N100, N45 and P60 amplitude during pain (see supplementary material).”

The supporting statistics for these analyses have been included in the supplementary material.

DISCUSSIONLine 303: " The present study determined whether acute experimental pain induces alterations in cortical inhibitory and/or facilitatory activity observed in TMS-evoked potentials".Well, no. The study assessed the N45, and was based on it. It did not really explore other metrics in a systematic fashion. P60 and N100 changes were not replicated in experiments 2 and 3..

We assume the reviewer is stating that we did not assess other TEP peaks (such as the N15, P30 and P180). However, we did indeed assess these peaks in a systematic fashion. First, we identified the ROI by using a cluster-based analysis. This is a recommended approach when the ROI is unclear (Frömer, Maier, & Abdel Rahman, 2018). We then analysed the TEP representing the mean voltage across the electrodes within the cluster, and then identified any differences in all peaks between conditions (not just the N45). This has been made clearer in the manuscript.

This has now been included:

“For all experiments, the mean TEP waveform of any identified clusters from Experiment 1 were plotted, and peaks (e.g., N15, P30, N45, P60, N100) were identified using the TESA peak function (Rogasch et al., 2017)”

Frömer, R., Maier, M., & Abdel Rahman, R. (2018). Group-level EEG-processing pipeline for flexible single trial-based analyses including linear mixed models. Frontiers in neuroscience, 12, 48.

And the N45 is not related to facilitatory or inhibitory activity, it is a measure of an evoked response indicating excitability

Evidence suggests the N45 is mediated by GABAAergic neurotransmission (inhibitory activity), as drugs which increase GABAA receptor activity increase the amplitude of the N45 (Premoli et al., 2014) and drugs which decrease GABAA receptor activity decrease the amplitude of the N45 (Darmani et al., 2016). As such, we and various other empirical papers (e.g., Bellardinelli et al., 2021; Noda et al., 2021; Opie at 2019 ) and review papers (Farzan & Bortoletto, 2022; Tremblay et al., 2019) have interpreted changes in the N45 peak as reflecting changes in cortical inhibitory/GABAA mediated activity.

Premoli, I., Castellanos, N., Rivolta, D., Belardinelli, P., Bajo, R., Zipser, C., ... & Ziemann, U. (2014). TMS-EEG signatures of GABAergic neurotransmission in the human cortex. Journal of Neuroscience, 34(16), 5603-5612.

Belardinelli, P., König, F., Liang, C., Premoli, I., Desideri, D., Müller-Dahlhaus, F., ... & Ziemann, U. (2021). TMS-EEG signatures of glutamatergic neurotransmission in human cortex. Scientific reports, 11(1), 8159.

Darmani, G., Zipser, C. M., Böhmer, G. M., Deschet, K., Müller-Dahlhaus, F., Belardinelli, P., ... & Ziemann, U. (2016). Effects of the selective α5-GABAAR antagonist S44819 on excitability in the human brain: a TMS–EMG and TMS–EEG phase I study. Journal of Neuroscience, 36(49), 12312-12320.

Noda, Y., Barr, M. S., Zomorrodi, R., Cash, R. F., Lioumis, P., Chen, R., ... & Blumberger, D. M. (2021). Single-pulse transcranial magnetic stimulation-evoked potential amplitudes and latencies in the motor and dorsolateral prefrontal cortex among young, older healthy participants, and schizophrenia patients. Journal of Personalized Medicine, 11(1), 54.

Farzan, F., & Bortoletto, M. (2022). Identification and verification of a'true'TMS evoked potential in TMS-EEG. Journal of neuroscience methods, 378, 109651.

Opie, G. M., Foo, N., Killington, M., Ridding, M. C., & Semmler, J. G. (2019). Transcranial magnetic stimulation-electroencephalography measures of cortical neuroplasticity are altered after mild traumatic brain injury. Journal of Neurotrauma, 36(19), 2774-2784.

Tremblay, S., Rogasch, N. C., Premoli, I., Blumberger, D. M., Casarotto, S., Chen, R., ... & Daskalakis, Z. J. (2019). Clinical utility and prospective of TMS–EEG. Clinical Neurophysiology, 130(5), 802-844.

Line 321: why have you not measured SEPs in experiment 3?

It is not possible to directly measure the somatosensory evoked potentials resulting from a TMS pulse, given that the TMS pulse produces a range of signals including cortical activity, muscle/eye blink responses, auditory responses, somatosensory responses and other artefacts. While some researchers attempt to isolate the SEP from TMS using pre-processing methods such as ICA, others use control conditions such as sensory sham conditions (to control for the “tapping” artefact) or subthreshold intensity conditions (to control for reafferent muscle activity), as we have done in Experiment 2 and 3 of our study.

We have now stated this in the manuscript:

“As it is extremely challenging to isolate and filter these auditory and somatosensory evoked potentials using pre-processing pipelines, masking methods have been used to suppress these sensory inputs, (Ilmoniemi and Kičić, 2010; Massimini et al., 2005). However recent studies have shown that even when these methods are used, sensory contamination of TEPs is still present, as shown by commonalities in the signal between active and sensory sham conditions that mimic the auditory/somatosensory aspects of real TMS (Biabani et al., 2019; Conde et al., 2019; Rocchi et al., 2021). This has led many leading authors (Biabani et al., 2019; Conde et al., 2019) to recommend the use of sham conditions to control for sensory contamination”

Line 365: SICI is dependent on GABAa activity. But the way the text is written if conveys the idea that TMS pulses "activate" GABA receptors, which is weird...Please rephrase.

This has now been reworded.

“SICI refers to the reduction in MEP amplitude to a TMS pulse that is preceded 1-5ms by a subthreshold pulse, with this reduction believed to be mediated by GABAA neurotransmission (Chowdhury et al., 2022)”

**Reviewer #3 (Recommendations For The Authors):**
-Key references Ye et al., 2022 and Che et al., 2019 need to be included in the reference list.

These references have now been included in the reference list.

-Heat pain stimuli and TMS stimuli are applied simultaneously. Sometimes the term "stimulus" is used without specifying whether it refers to TMS pulses or heat pain stimuli. Clarifying this whenever the word "stimulus" is used would enhance clarity for the reader.

We have now clarified the use of the word “stimulus” throughout the paper.

-Panels A-D in Figure 6 should be correctly labeled in the text and the figure legend.

Figure 6 Panel labels have now been amended.